# Scalable Reinforcement Learning via Adaptive Batch Scaling

**Jongchan Park** [* 1]

## Abstract

Conventional wisdom holds that large-batch training is fundamentally incompatible with Reinforcement Learning (RL) — beyond a modest threshold, increasing batch sizes typically yields diminishing returns or performance degradation due to the inherent non-stationarity of the data distribution. We challenge this view by observing that non-stationarity is not a fixed property of RL, but evolves throughout training: early stages exhibit rapid behavioral shifts that demand small batches for plasticity, whereas late stages approach a quasi-stationary regime where large batches enable precise convergence. Motivated by this observation, we propose Adaptive Batch Scaling (ABS), that dynamically adjusts the effective batch size according to the stability of the learning policy. Central to ABS is Behavioral Divergence, a novel metric that quantifies policy non-stationarity by measuring action-level shifts between consecutive updates, which we use to scale batch size inversely to policy volatility. Integrated with the Parallelised Q-Network (PQN) algorithm and evaluated on the ALE benchmark, ABS seamlessly reconciles early-stage plasticity with late-stage stable convergence. Strikingly, contrary to conventional wisdom, our results reveal that the combination of larger networks and larger batch sizes achieves the best performance — a scaling behavior previously thought to be unattainable in RL, now unlocked through adaptive batch control. Our code is available at https://github.com/daisophila/ABS.

## 1. Introduction

The prevailing paradigm in modern Deep Learning is defined by the 'Scaling Laws', where performance monotonically improves with increased model size, dataset size, and computational budget (Kaplan et al., 2020). In domains such as Computer Vision (CV) and Natural Language Processing (NLP), this has crystallized into a strategy of training massive models with correspondingly massive batch sizes. Large batch training not only stabilizes the gradient estimation by reducing variance but also maximizes parallelization efficiency on distributed hardware accelerators. Pioneering works in large-scale optimization, such as LARS (You et al., 2017) and LAMB (You et al., 2019), have enabled training ResNet-50 on ImageNet in minutes using batch sizes of up to 32k. Similarly, the success of Large Language Models (LLMs) like GPT-3, DeekSeek-R1 (Brown et al., 2020; Guo et al., 2025) and Vision Transformers (ViT) (Dosovitskiy, 2020; Zhai et al., 2022) relies heavily on global batch sizes comprising millions of tokens to effectively saturate compute capacity and maintain training stability.

However, Reinforcement Learning (RL) presents a stark contrast to this trend. Despite the adoption of scalable architectures like Transformers, the effective batch sizes in RL remain surprisingly small compared to their supervised counterparts. For instance, the Distributed Impala architecture (Espeholt et al., 2018) and even high-capacity Robotics Transformers (RT-1, RT-2) (Brohan et al., 2022; Zitkovich et al., 2023) operate on surprisingly modest batch sizes (e.g., 256 to 1024) that do not scale in proportion to their massive parameter counts. This phenomenon extends to Large Language Model (LLM) training, where reinforcement learning stages utilize notably smaller batch sizes compared to the massive scales employed during supervised learning phases (Rafailov et al., 2023; Guo et al., 2025). Empirical studies consistently report that increasing the batch size beyond a critical threshold in RL often leads to diminishing returns or even performance degradation (Obando Ceron et al., 2023; McCandlish et al., 2018).

We attribute this phenomenon to the inherent non-stationarity of both the evolving policy and the data distribution generated through environment interactions, and it creates a fundamental bias-variance tradeoff. Unlike supervised learning, which operates on a fixed data distribution, RL agents must learn from a stream of experience that co-evolves with the policy updates ($\pi_\theta$). Recent studies have highlighted the fundamental difficulty of optimizing deep models under such shifting distributions, even in supervised

[1]Hyundai Motor Company. Correspondence to: Jongchan Park <jcpark11@hyundai.com>.

*Proceedings of the 43rd International Conference on Machine Learning*, Seoul, South Korea. PMLR 306, 2026. Copyright 2026 by the author(s).

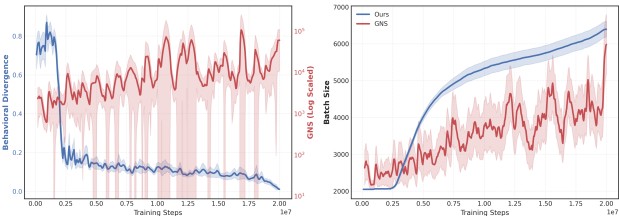

*Figure 1.* **Evolution of Behavioral Divergence, Gradient Noise Scale (GNS) and Batch Size Scaling. (Left)** Behavioral divergence and GNS aggregated across 10 Atari environments (3 seeds each). As the policy approaches a near-stationary stage, the divergence diminishes while the GNS increases. **(Right)** Correspondingly, ABS dynamically scales up the batch size, responding to the increased stability of the learning process.

contexts (Castanyer et al., 2025). This challenge is particularly acute during the initial stages of RL training, where rapid policy shifts exacerbate distributional instability. To mitigate these effects, employing modest batch sizes is often preferred, as it allows the agent to adapt more fluidly to the non-stationary data stream (Obando Ceron et al., 2023).

This paper challenges the static view of batch size in RL based on a simple yet overlooked insight: the degree of non-stationarity is not constant throughout training. Figure 1 (Left, Blue curve) illustrates the evolution of the policy's behavior change rate between successive updates (*i.e., defined 'Behavioral Divergence' in this work*)—over aggregated Atari-10 (Aitchison et al., 2023) environments. In the initial phase, intensive exploration and rapid parameter adjustments lead to high behavioral volatility, resulting in severe distribution shifts. In this highly non-stationary regime, maintaining small batch sizes (*i.e., short rollouts for RL*) is essential to preserve model plasticity (*i.e, rapidly evolving policies*) for minimizing bias and prevent the agent from overfitting to transient, unstable trajectories. However, as training progresses and the policy converges toward optimal behavior, the rate of behavioral change diminishes significantly, and the trajectory distribution approaches a near-stationary state. In this 'near-stationary' regime, large batches (*i.e., long rollouts for RL*) can help suppress gradient noise and ensure convergence by minimizing variance. We hypothesize that adhering to small batches in this later stage is suboptimal; instead, scaling the batch size—as per standard Deep Learning wisdom—becomes necessary to refine the policy with high-precision gradients and accelerate convergence.

Based on this hypothesis, we propose a novel simple yet effective training framework that dynamically adjusts the batch size (*i.e., the rollout length of an RL*) in response to the stability of the learning process. Specifically, we introduce a mechanism that scales the batch size inversely proportional to the policy behavioral change rate (Figure 1 (Left)). We demonstrate the efficacy of this approach by

applying it to PQN (Gallici et al., 2024) on the ALE benchmark (Bellemare et al., 2013). Empirical results demonstrate that our Adaptive Batch Scaling (ABS) yields substantial performance improvements by synergizing early-stage stability with high-throughput optimization in the final phases. Notably, this approach effectively scales to larger backbone architectures (Castanyer et al., 2025), surmounting the performance bottlenecks inherent in conventional RL training. Given its robust performance and ease of integration, we expect our methodology to be a foundational key to unlocking the scaling potential of reinforcement learning, bridging the performance gap with supervised learning at large scale architectures.

Our contributions are summarized as follows:

- **Quantification of RL Non-stationarity:** We introduce **Behavioral Divergence**—a novel metric that quantifies the non-stationarity of RL training by measuring action-level shifts between successive policy updates.

- **Adaptive Batch Scaling Framework:** We propose a novel simple yet effective algorithm that dynamically scales the batch size based on policy stability metrics (*i.e.*, Behavioral Divergence).

- **Scalability Verification:** We empirically demonstrate that our method enables performance improvements with larger batch sizes on the ALE benchmark, particularly when combined with larger network architectures.

## 2. Related Work

**Batch Size Scalability in Reinforcement Learning.** Recent endeavors in Reinforcement Learning (RL) have sought to replicate the 'Scaling Laws' of supervised learning by scaling up model parameters and architectures (Espeholt et al., 2018; Brohan et al., 2022; Zitkovich et al., 2023; Wang et al., 2025; Castanyer et al., 2025). However, unlike model size, the batch size—a critical hyperparameter for training speed and stability—has remained disproportionately small. This discrepancy is particularly evident in the RL fine-tuning stages of Large Language Models (LLMs); while Pretraining and Supervised Fine-Tuning (SFT) utilize massive batch sizes, the subsequent RL phases (e.g., RLHF) revert to significantly smaller batches (Rafailov et al., 2023; Guo et al., 2025; Yu et al., 2025). Empirical studies suggest that smaller batch sizes generally yield better performance in standard RL benchmarks due to the inherent non-stationarity of the learning process (Obando Ceron et al., 2023; McCandlish et al., 2018). To overcome this, frequent updates with small batches are typically employed to track the shifting data distribution.

In this work, we view RL training through the lens of two distinct regimes: the early and late phases. The early phase

is characterized by high policy volatility and significant non-stationarity, necessitating small batches for rapid adaptation. Conversely, as training progresses, the policy stabilizes, and the data distribution becomes 'near-stationary', resembling a supervised learning task. We propose a methodology that bridges these regimes by starting with small batches to mitigate early-stage non-stationarity and progressively scaling them up to enhance stability and convergence speed in the later stages.

**Dynamic Batch and Sampling Mechanisms.** Prior research has explored dynamically adjusting batch size to improve efficiency (Bhatia et al., 2022; Yu et al., 2025; McCandlish et al., 2018). In Model-Based RL, (Bhatia et al., 2022) proposed dynamically adjusting the rollout length to balance the trade-off between sample efficiency and model accuracy. In the context of data selection, (Yu et al., 2025) maintained a fixed batch size but adjusted the number of candidate samples to ensure each batch contained high-information data (DAPO).

Most relevant to our work is (McCandlish et al., 2018), which introduced the Gradient Noise Scale (GNS) to determine the optimal batch size at any given step. However, leveraging GNS directly for online RL training presents practical challenges: it requires additional backward passes to compute per-sample gradients, incurring significant computational overhead. Furthermore, while their analysis showed that the optimal batch size (inferred from increasing GNS over steps) increases over time, they did not experimentally validate a dynamic batch sizing schedule in RL, instead concluding that fixed small batches were empirically superior for the tested RL tasks—Figure 1 is the result that expands the GNS based batch scheduling to RL by us. Our work aligns with the theoretical implication of (McCandlish et al., 2018)—that batch size should grow—but diverges in execution. We propose a metric based on the policy action change rate (Behavioral Divergence), which relies solely on the forward pass, making it computationally efficient. We demonstrate that this lightweight, dynamic batch size scheduling strategy effectively handles the stationarity transition and improves performance across various ALE environments.

**Policy Churn and Behavioral Divergence.** The instability of greedy policies during training has been studied under the notion of *policy churn* (Schaul et al., 2022; Tang & Berseth, 2024). Schaul et al. (2022) first documented that a single gradient update can alter the greedy actions across a substantial fraction of states, hypothesizing that this rapid fluctuation serves as a mechanism for *implicit exploration*. Tang & Berseth (2024) further characterized a *chain effect* in which value churn and policy churn compound cyclically, leading to training instability, and proposed a regularization method (CHAIN) to suppress these oscillations and stabilize learning.

Our notion of *Behavioral Divergence* is conceptually related to policy churn in that both quantify the discrepancy between successive policies. However, the objectives diverge fundamentally: prior work treats churn as a phenomenon to be *explained* (as implicit exploration) or *mitigated* (via regularization), whereas we leverage Behavioral Divergence as an *metric* for non-stationarity in the learning process.

# 3. Preliminaries

We consider a standard Markov Decision Process (MDP) defined by the tuple $(\mathcal{S}, \mathcal{A}, P, R, \gamma)$, where $\mathcal{S}$ and $\mathcal{A}$ denote the state and action spaces, $P$ the transition dynamics, $R$ the reward function, and $\gamma \in [0, 1)$ the discount factor. The objective of the agent is to learn a parameterized policy $\pi_\theta(a|s)$ that maximizes the expected cumulative reward $J(\theta) = \mathbb{E}_{\tau \sim \pi_\theta}[\sum_{t=0}^{\infty} \gamma^t R(s_t, a_t)]$.

## 3.1. Bias-Variance in RL

In modern on-policy style algorithms such as PPO (Schulman et al., 2017), the policy gradient is typically estimated using a batch of trajectories $\mathcal{B}$ collected by the behavior policy $\pi_{\theta_{\text{old}}}$. The objective gradient is approximated as:

$$\hat{\nabla}_\theta J(\theta) \approx \frac{1}{|\mathcal{B}|} \sum_{(s,a)\in\mathcal{B}} \frac{\pi_\theta(a|s)}{\pi_{\theta_{\text{old}}}(a|s)} \hat{A}(s,a) \cdot \nabla_\theta \log \pi_\theta(a|s)$$

(1)

where $\hat{A}$ represents the advantage estimator (e.g., GAE). The quality of this gradient estimate is governed by the bias-variance tradeoff, which is directly influenced by the batch size $B = |\mathcal{B}|$.

**Variance (Estimation Error).** The variance of the gradient estimator arises from the stochasticity of the environment (transitions and rewards) and the policy itself. The variance of the gradient estimate scales inversely with the batch size:

$$\text{Var}(\hat{\nabla} J) \propto \frac{1}{B}$$

(2)

Small batches ($B \downarrow$) yield high variance results in 'noisy' gradients. While traditionally seen as detrimental, this stochastic noise acts as a source of implicit exploration, preventing the policy from prematurely converging to sharp local minima. In contrast, larger batches ($B \uparrow$) average out the noise, providing a high-fidelity estimate of the true gradient direction (Smith et al., 2017).

**Bias (Approximation & Shift Error).** In RL, bias primarily stems from two sources: (1) the bias in advantage estimation (e.g., due to bootstrapping in GAE), and (2) the

*distributional shift* during multi-epoch updates. When optimizing $\theta$ over multiple epochs using a fixed batch $\mathcal{B}$, the data distribution $d^{\pi_\theta}$ diverges from the collection distribution $d^{\pi_{\theta_{\text{old}}}}$.

While large batches provide a stable signal for the *collected* distribution, they encourage the optimizer to commit strongly to a specific policy update direction. If the underlying data distribution is non-stationary (i.e., the optimal policy changes drastically), 'over-optimizing' on a large batch introduces a bias where the agent overfits to transient dynamics that may no longer be valid after the update.

### 3.2. The Dynamics of Non-stationarity: Early vs. Late Training

The central premise of our work is that the optimal trade-off between bias and variance is not static but evolves as training progresses.

**Early Stage: High Non-stationarity & Plasticity.** At the beginning of training, the policy $\pi_\theta$ is initialized randomly, leading to high behavioral volatility. Consequently, the distribution shift $d^{\pi_k} \rightarrow d^{\pi_{k+1}}$ is drastic between updates. In this regime, the learning landscape is highly non-stationary.

Calculating a precise gradient (Large Batch) for a transient policy is computationally wasteful and potentially harmful. The 'accurate' gradient may point towards a local optimum of a distribution that is about to vanish. Paradoxically, the high variance from **small batches acts as an implicit regularizer, preserving *model plasticity* and preventing the agent from overfitting to unstable trajectories** (Igl et al., 2020).

**Late Stage: Near-Stationarity & Convergence.** As the agent begins to master the task, the policy $\pi_\theta$ converges towards a high-performing region, and the rate of behavioral change diminishes. The trajectory distribution approaches a *quasi-stationary* state.

In this stable regime, the gradient noise from small batches transitions from a feature (exploration) to a bug (instability), preventing the model from settling into the sharp global optimum. Here, **scaling up the batch size reduces variance, allowing for high-precision updates required to refine the policy and accelerate asymptotic convergence** (Smith et al., 2017).

### 3.3. Parallelised Q-Network (PQN)

In our experiments, we utilize the Parallelised Q-Network (PQN) proposed by (Gallici et al., 2024) as the primary learning framework. PQN simplifies deep temporal difference (TD) learning by integrating the stability benefits of proximal optimization into value-based methods. Specifi-

---

**Algorithm 1** PQN with Adaptive Batch Scaling (ABS)

**Input:** Initial parameters $\theta$, environment count $E$, update frequency $K$, thresholds $\delta_{\min}, \delta_{\max}$, bounds $L_{\min}, L_{\max}$.

**while** $t < T_{\text{total}}$ **do**
  If $t \pmod{K} == 0$, set $\theta_{\text{old}} \leftarrow \theta$.

  **Step 1: Data Collection**
  Collect trajectories for $L_{\text{adapt}}$ steps using $E$ environments and policy $\pi_\theta$.
  $t = t + (E \times L_{\text{adapt}})$

  **Step 2: Policy Update**
  **for** $epoch = 1$ **to** $N_{\text{epochs}}$ **do**
    Sample mini-batches from the collected trajectories.
    Update $\theta$ by minimizing PQN loss.
  **end for**

  **Step 3: Adaptive Scaling**
  **if** $t \pmod{K} == 0$ **then**
    Sample reference batch $\mathcal{B}_{\text{ref}}$ of size $M$ from the collected trajectories.
    Calculate behavioral divergence:
      $\delta_\pi = \frac{1}{M} \sum \mathbb{I}[\pi_\theta(s) \neq \pi_{\theta_{\text{old}}}(s)], \quad s \sim \mathcal{B}_{\text{ref}}$
    Calculate $L'_{\text{adapt}}$ based on $\delta_\pi$ (Eq. 4).
    Smooth update: $L_{\text{adapt}} \leftarrow (1-\alpha)L_{\text{adapt}} + \alpha L'_{\text{adapt}}$
  **end if**
**end while**

---

cally, instead of relying on slowly-updating target networks, PQN employs a proximal objective that clips the Q-value updates relative to the previous estimates, preventing catastrophic divergence during frequent gradient steps. By applying our Adaptive Batch Scaling (ABS) to PQN, we aim to demonstrate that even highly efficient, modern RL architectures can further benefit from dynamic batching, particularly when scaling to larger neural backbones.

## 4. Method

In this section, we introduce **Adaptive Batch Scaling (ABS)**, a training framework designed to align the batch size with the evolving stationarity of the reinforcement learning process. Our core insight is to utilize small batches during the highly non-stationary early training phase to maximize gradient update frequency and plasticity, while transitioning to larger batches in the stationary later phase to ensure stable convergence. We quantify stationarity via the rate of policy behavioral change (*i.e.*, behavioral divergence) and employ this metric to dynamically adjust the data collection horizon (rollout length). We demonstrate the implementation of ABS within the Parallelised Q-Network (PQN) algorithm.

### 4.1. Quantifying Non-stationarity via Behavioral Divergence

In RL, the learning agent alternates between data collection using the current policy $\pi_\theta$ and optimization updates. Unlike the fixed data distribution in supervised learning, the distribution of collected experience in RL is non-stationary, shifting continuously as the policy evolves. While the environment dynamics $P(s'|s,a)$ generally remain static, the policy distribution $\pi_\theta(a|s)$ induces a covariate shift in the visited state-action space.

This distributional shift is most severe during the early stages of training when policy updates result in significant behavioral changes. As training progresses and the policy converges toward an optimal behavior, the updates become incremental, leading to a 'near-stationary' joint distribution of the policy and environment.

To operationalize this observation, we propose **Behavioral Divergence** ($\delta_\pi$) as a proxy metric for measuring non-stationarity. Specifically, we measure the rate at which the updated policy $\pi_{\theta'}$ deviates from a rollout policy $\pi_\theta$. Formally, $\delta_\pi$ is defined as the fraction of states where the preferred actions differ:

$$\delta_\pi(\theta', \theta) := \frac{1}{M} \sum_{i=1}^{M} \mathbb{I}\left[\pi_{\theta'}(s_i) \neq \pi_\theta(s_i)\right], \; s_i \sim \mathcal{B}_{ref} \quad (3)$$

where $\mathcal{B}_{ref}$ is a reference batch of $M$ states sampled from recent trajectories (*i.e.*, collected by $\pi_\theta$, $M = 2048$), and $\mathbb{I}[\cdot]$ is the indicator function. Since we build upon PQN, the policy is deterministically derived from the Q-values: $\pi_\theta(s) = \arg\max_a Q_\theta(s,a)$. As illustrated in Figure 1, $\delta_\pi$ typically starts high due to random exploration and rapid initial learning, then monotonically decreases as the agent stabilizes.

### 4.2. Adaptive Batch Scaling (ABS)

Standard RL algorithms which don't use replay buffer for learning define the global batch size $|\mathcal{B}|$ as the product of the number of parallel environments ($E$) and the rollout length ($L$): $|\mathcal{B}| = E \times L$. While $E$ is typically constrained by hardware resources, $L$ is a flexible hyperparameter that directly dictates the batch size and the freshness of the data.

ABS dynamically scales $|\mathcal{B}|$ by adjusting the rollout length $L$ in response to the measured Behavioral Divergence $\delta_\pi$. We maintain a fixed number of mini-batches $N_{mb}$ and optimization epochs $N_{epochs}$.

We define a scaling schedule where the rollout length $L_{adapt}$ increases as the divergence $\delta_\pi$ decreases. To bridge the varying scales of rollout lengths smoothly, we employ a

logarithmic interpolation between a minimum length $L_{min}$ and a maximum length $L_{max}$:

$$L_{adapt} = L_{max} - \alpha \cdot (L_{max} - L_{min}) \quad (4)$$

where the interpolation factor $\alpha$ is determined by the normalized divergence:

$$\alpha = \frac{\log(\delta_{\pi,\text{cliped}}/\delta_{min})}{\log(\delta_{max}/\delta_{min})} \quad (5)$$

Here, $\delta_{\pi,\text{cliped}} = \text{clip}(\delta_\pi, \delta_{min}, \delta_{max})$ and $[\delta_{min}, \delta_{max}]$ represents the sensitivity range of the policy change rate. If the policy is changing rapidly ($\delta_\pi \geq \delta_{max}$), we use $L_{min}$ to prioritize frequent updates. Conversely, if the policy is stable ($\delta_\pi \leq \delta_{min}$), we scale to $L_{max}$ to minimize gradient noise.

Finally, the effective batch size for the next update is determined as:

$$|\mathcal{B}| = E \cdot L_{adapt} \quad (6)$$

This allows the effective mini-batch size used for gradient descent to be $|\mathcal{B}|/N_{mb}$. To balance computational overhead and enhance the learning stability, we re-calculate $L_{adapt}$ every $K$ updates.

### 4.3. Implementation on PQN

We integrate our Adaptive Batch Scaling (ABS) framework into the Parallelised Q-Network (PQN) (Gallici et al., 2024), a prominent recent off-policy RL algorithm. The complete training procedure is summarized in Algorithm 1.

Our implementation is built upon the CleanRL library[1]. We uploaded our implementation of ABS on our github page[2] for reproducibility. Further implementation details and specific hyperparameter settings are provided in Appendix A and Appendix C, respectively.

## 5. Experiments

In this section, we empirically validate the effectiveness of our Adaptive Batch Scaling (ABS). Our experiments are designed to answer the following research questions:

- *Q1 (Performance). Does ABS improve performance compared to standard fixed-batch baselines and a prior dynamic batch approach?*

- *Q2 (Scalability). Can ABS stabilize the training and increase the performance of larger network architectures with scaled batch?*

---

[1] https://github.com/vwxyzjn/cleanrl
[2] https://github.com/daisophila/ABS

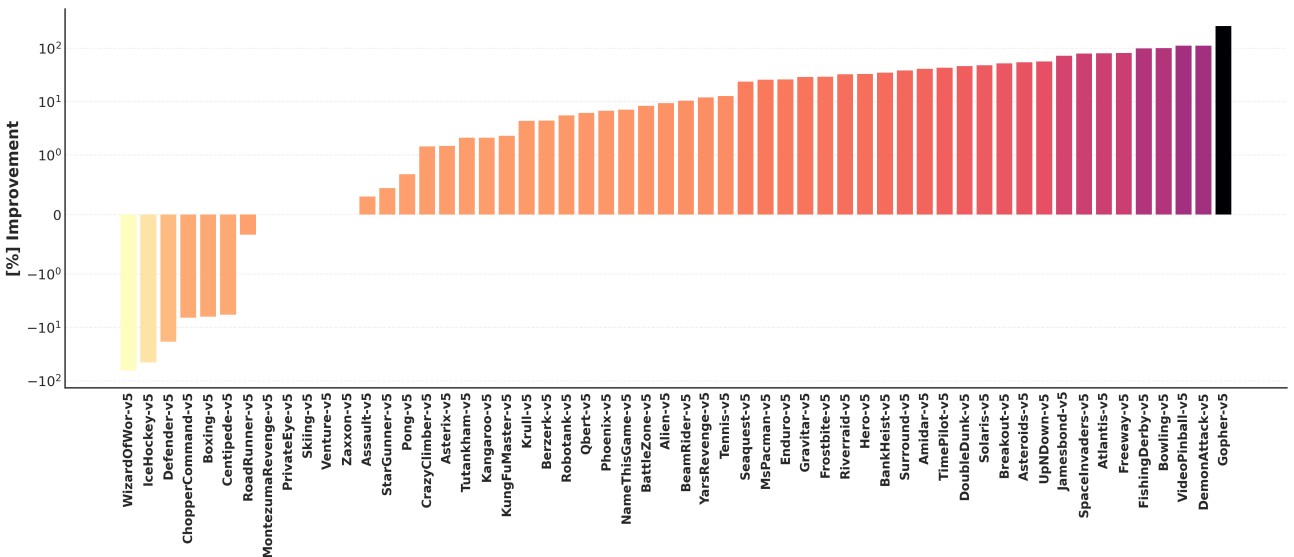

*Figure 2.* **Log Scaled Performance Improvement of PQN with ABS** relative to vanilla PQN on the Full ALE benchmark. Notably, ABS improves PQN performance in most environments.

- **Q3 (Sensitivity).** *How sensitive is ABS to its hyperparameters, and which 54components are most critical to its performance?*

- **Q4 (Generalization).** *Is ABS applicable to continuous control domains (e.g., PPO) and even to a recent off-policy algorithm (e.g., BTR) which involves a replay buffer ?*

Subsequence is the experimental setup to answer the above questions:

**Environments.** We evaluate our method on the full Arcade Learning Environment (ALE) (Bellemare et al., 2013), utilizing the Atari-57 benchmark. For ablation studies, we use the **Atari-10** suite (Aitchison et al., 2023), a curated subset of 10 games identified to capture the maximum variance in algorithm performance and highly correlated ($> 0.9$) with the full benchmark. Additionally, we extend our evaluation to continuous control tasks using **MuJoCo** environments via Gymnasium.

**Baselines & Implementation.** Our primary baseline is **PQN** (Gallici et al., 2024), a streamlined and high-throughput but effective RL algorithm. We adapt our ABS on PQN and compare to: (1) vanilla PQN, (2) PQN with smaller fixed batch size, (3) PQN with larger fixed batch size, and (4) PQN with Gradient Noise Scale (GNS) (McCandlish et al., 2018) based dynamic batch scheduling. For scalability experiments, we adopt the ResNet-style multi-skip architectures proposed in (Castanyer et al., 2025) (*i.e.*, but using same optimizer with PQN, RAdam (Liu et al., 2019)).

**Evaluation Protocol.** We train all agents for 20 million steps (5M for MuJoCo) across 3 independent seeds following standard protocols (Agarwal et al., 2021; Obando-Ceron et al., 2024). For evaluation, we report the mean score over 100 evaluation episodes per seed (total 300 episodes) after training concludes.

### 5.1. *A1: Enhancing Performance on Full Atari-57*

We first evaluate ABS on the full Atari-57 benchmark. Figure 2 illustrates the relative performance gains of ABS compared to the vanilla PQN baseline, showing that our method outperforms the baseline across the most of environments. To further analyze the training dynamics, Figure 3 presents the learning curves on Atari-10, averaged over three seeds.

Table 4 shows early training performance (first 10% of steps) and late training stability (last 10% of steps) separately. ABS achieves higher early-stage scores in 7/10 environments, supporting the hypothesis that smaller batches accelerate initial learning. For late-stage variance, ABS achieves lower variance in 6/10 environments. These results validate our hypothesis: leveraging smaller batch sizes in the early stages accelerates initial learning, while transitioning to larger batches in the later stages stabilizes and enhances final performance.

### 5.2. *A1: Outperforming Fixed Batches*

To isolate the impact of batch size, we compare ABS against fixed Small Batch and Large Batch configurations on the Atari-10 suite. The Small Batch setting utilizes a rollout length of 16 (2,048 samples), while the Large Batch setting

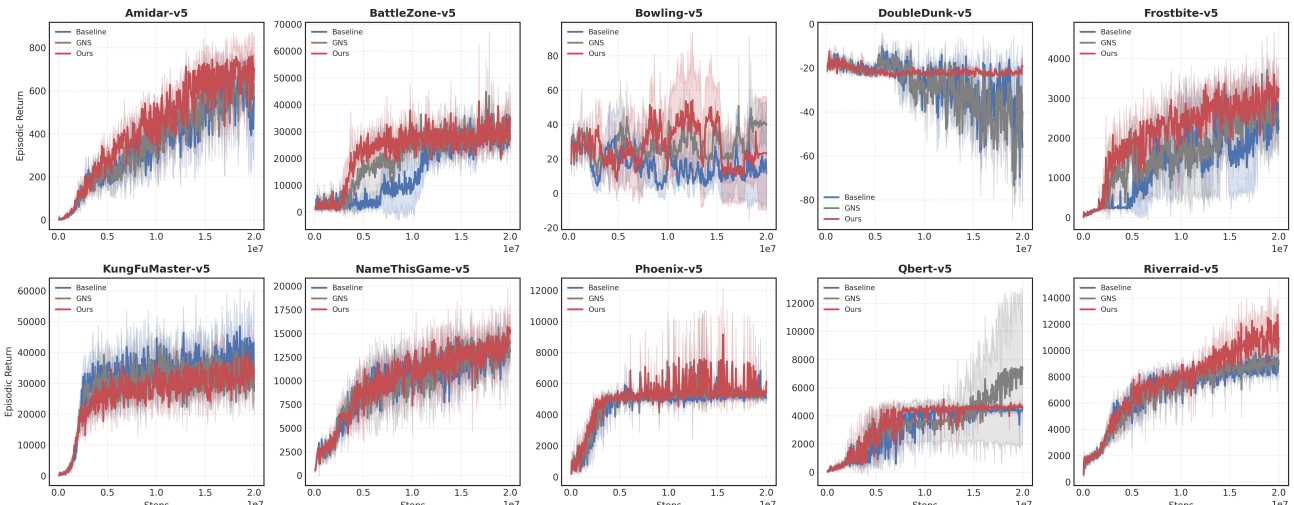

*Figure 3.* **Learning Curves of Vanilla PQN, with ABS(Ours) and GNS** on Atari-10. ABS (ours) boosts the overall performance of PQN across most tasks, surpassing both vanilla PQN and GNS in efficiency.

*Table 1.* **Fixed and Dynamic Batch Sizes on Atari-10.** The results show that ABS outperforms over all of baselines.

| Environment | PQN | Small Batch | Large Batch | GNS | PQN + ABS(Ours) |
|---|---|---|---|---|---|
| Amidar-v5 | $0.32 \pm 0.12$ | $0.42 \pm 0.14$ | $0.38 \pm 0.04$ | $0.37 \pm 0.09$ | **$0.43 \pm 0.07$** |
| BattleZone-v5 | $0.80 \pm 0.12$ | $0.76 \pm 0.04$ | $0.68 \pm 0.11$ | $0.78 \pm 0.26$ | **$0.85 \pm 0.07$** |
| Bowling-v5 | $-0.13 \pm 0.12$ | $-0.03 \pm 0.10$ | **$0.25 \pm 0.14$** | $0.10 \pm 0.10$ | $-0.08 \pm 0.25$ |
| DoubleDunk-v5 | $-9.80 \pm 5.70$ | $-1.94 \pm 0.09$ | $-1.56 \pm 0.68$ | $-12.98 \pm 2.64$ | **$-1.44 \pm 0.57$** |
| Frostbite-v5 | $0.48 \pm 0.13$ | **$0.90 \pm 0.08$** | $0.49 \pm 0.20$ | $0.54 \pm 0.19$ | $0.67 \pm 0.15$ |
| KungFuMaster-v5 | $1.56 \pm 0.06$ | **$1.61 \pm 0.11$** | $1.43 \pm 0.30$ | $1.42 \pm 0.17$ | $1.57 \pm 0.20$ |
| NameThisGame-v5 | $1.92 \pm 0.05$ | $1.95 \pm 0.37$ | $2.04 \pm 0.10$ | $1.75 \pm 0.32$ | **$2.05 \pm 0.06$** |
| Phoenix-v5 | $0.66 \pm 0.02$ | $0.70 \pm 0.03$ | $0.69 \pm 0.01$ | **$0.75 \pm 0.02$** | $0.71 \pm 0.02$ |
| Qbert-v5 | $0.31 \pm 0.01$ | $0.33 \pm 0.02$ | $0.33 \pm 0.03$ | **$0.79 \pm 0.31$** | $0.34 \pm 0.01$ |
| Riverraid-v5 | $0.45 \pm 0.02$ | $0.51 \pm 0.05$ | $0.54 \pm 0.21$ | $0.49 \pm 0.02$ | **$0.64 \pm 0.12$** |

uses a rollout length of 64 (8,192 samples), compared to the default PQN's 32 (4,096 samples). Table 1 summarizes these results with IQM HNS ± 95% confidence intervals. While Small Batch often outperforms the vanilla baseline by prioritizing plasticity and rapid updates, ABS achieves the superior score in 8 out of 10 environments. This success stems from its ability to resolve the fundamental trade-off between these regimes: ABS leverages small batches to navigate early-stage instabilities and maintain plasticity, then scales to large batches to minimize gradient variance for precise late-stage refinement.

### 5.3. *A1: Outperforming Gradient Noise Scale (GNS)*

We compare ABS with the Gradient Noise Scale (GNS) metric proposed by (McCandlish et al., 2018), which suggests batch sizes based on gradient variance. While GNS is theoretically grounded, it requires computationally expensive backward passes to estimate noise. In contrast, ABS relies solely on forward pass inference (behavioral divergence), making it significantly more efficient.

As shown in the learning curves (Figure 3) and evaluation results (Table 1), ABS outperforms GNS-based scheduling in terms of sample efficiency and final performance. This suggests that our *behavioral divergence* is a more direct and practical way for measuring non-stationarity and batch

*Table 2.* **Scaled PQN with a Large fixed Batch and Our ABS on Atari-10**. Our ABS outperforms over all of baseline whereas the large fixed batch decreases the performance.

| Environment | PQN-L (4,096, default) | PQN-L (16,384) | PQN-L + ABS (2,048→16,384) | PQN-XL (4,096, default) | PQN-XL (16,384) | PQN-XL + ABS (2,048→16,384) |
|---|---|---|---|---|---|---|
| Amidar-v5 | $0.39 \pm 0.04$ | $0.30 \pm 0.06$ | **$0.46 \pm 0.09$** | $0.41 \pm 0.02$ | $0.30 \pm 0.07$ | **$0.42 \pm 0.05$** |
| BattleZone-v5 | **$0.68 \pm 0.55$** | $0.56 \pm 0.27$ | $0.05 \pm 0.10$ | $-0.05 \pm 0.03$ | **$0.79 \pm 0.11$** | $0.51 \pm 0.38$ |
| Bowling-v5 | $-0.13 \pm 0.11$ | $-0.10 \pm 0.06$ | **$0.05 \pm 0.01$** | $0.01 \pm 0.12$ | $0.05 \pm 0.00$ | **$0.07 \pm 0.10$** |
| DoubleDunk-v5 | $-12.42 \pm 4.11$ | $-2.17 \pm 0.17$ | $-2.09 \pm 0.27$ | $-2.39 \pm 0.09$ | $-2.40 \pm 0.09$ | **$-1.79 \pm 0.34$** |
| Frostbite-v5 | $0.61 \pm 0.18$ | $0.59 \pm 0.34$ | **$1.76 \pm 0.43$** | $0.27 \pm 0.33$ | $0.84 \pm 0.41$ | **$1.69 \pm 0.68$** |
| KungFuMaster-v5 | $1.11 \pm 0.59$ | $0.88 \pm 0.19$ | **$1.88 \pm 0.22$** | **$1.88 \pm 0.37$** | $0.82 \pm 0.35$ | $0.89 \pm 0.24$ |
| NameThisGame-v5 | $0.42 \pm 0.35$ | $0.11 \pm 0.18$ | **$0.65 \pm 0.17$** | $0.25 \pm 0.17$ | $0.06 \pm 0.04$ | **$0.99 \pm 0.43$** |
| Phoenix-v5 | **$0.67 \pm 0.05$** | $0.67 \pm 0.02$ | $0.64 \pm 0.24$ | $0.81 \pm 1.00$ | $0.61 \pm 0.05$ | **$2.66 \pm 0.92$** |
| Qbert-v5 | $0.38 \pm 0.42$ | $0.19 \pm 0.12$ | **$0.50 \pm 0.31$** | $0.11 \pm 0.14$ | $0.34 \pm 0.41$ | **$0.37 \pm 0.26$** |
| Riverraid-v5 | $0.45 \pm 0.05$ | $0.37 \pm 0.10$ | **$0.48 \pm 0.04$** | $0.46 \pm 0.02$ | $0.39 \pm 0.04$ | **$0.47 \pm 0.03$** |

scheduling in RL than gradient noise alone. Details and hyperparameters of the GNS is in Appendix A.2 and C.

### 5.4. *A2: Unlocking the Potential of Scaling RL with ABS*

In general deep learning, larger models typically necessitate larger batch sizes to ensure convergence and maximize final performance (Hernandez & Brown, 2020). However, in standard RL, simply increasing batch size often leads to performance degradation on larger models. A central hypothesis of this work is that ABS can significantly enhance the performance of large-scale RL models by unlocking the potential of their vast parameter capacity through larger batch in late learning stage.

To verify this, we adopt ResNet-style Multi-Skip architectures from (Castanyer et al., 2025), specifically the PQN-L (CNN + 10-layer MLP) and PQN-XL (CNN + 20-layer MLP), for scaled large RL policy. Table 2 presents the IQM HNS with 95% confidence intervals on the Atari-10 benchmark. We find that ABS consistently and significantly enhances the performance of scaled networks, while a large fixed batch size conversely degrades performance across all model capacities.

Figure 4 presents the mean improvement rate of ABS

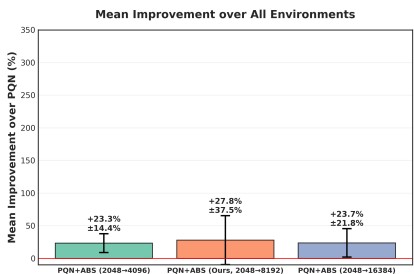 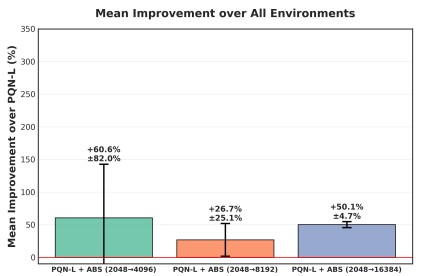 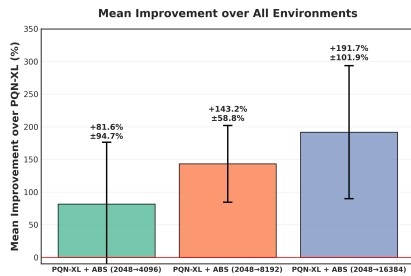

*Figure 4.* **Mean Improvement Rate of ABS over each Baseline** (Left: PQN, Center: PQN-L and Right: PQN-XL) on Atari-10, as a function of Rollout Range. The improvement rate consistently grows with model capacity and with larger $L_{max}$ (maximum rollout length; batch size), demonstrating a positive correlation between model scale and the efficacy of adaptive batch.

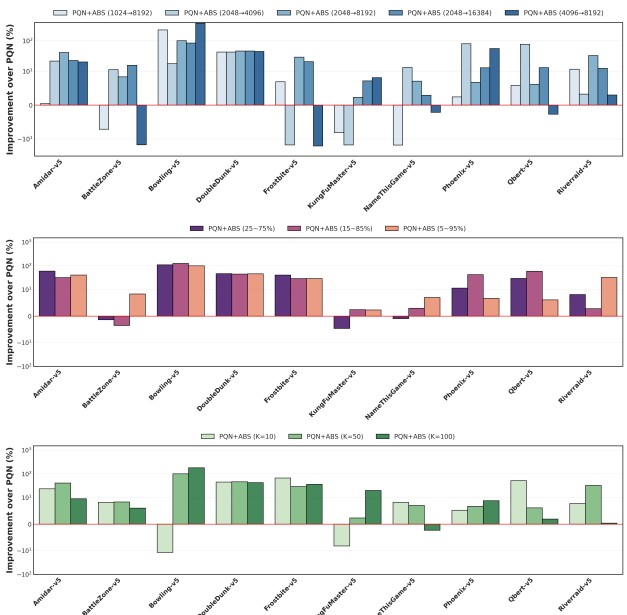

*Figure 5.* **Log Scaled Performance Improvement of PQN with ABS over various Hyperparameter settings** (Top: Rollout Range, Center: Policy Change Thresholds and Bottom: Adapt Frequency) on Atari-10.

over each baseline across the Atari-10 environments, organized into three bar plots corresponding to three model scales: PQN, PQN-L, and PQN-XL. Within each plot, the x-axis denotes the rollout range $[L_{min} \rightarrow L_{max}]$ (16→32; 2,048→4,096 samples, 16→64; 2,048→8,192 samples, 16→128; 2,048→16,384 samples), and the y-axis denotes the mean improvement rate (%) over the respective baseline.

As model capacity and maximum rollout length (batch size) increase, the improvement of ABS becomes more pronounced. For PQN-XL, the largest model, ABS (16→128; 2,048→16,384 samples) achieves the most substantial gains. For detailed results, please refer to Appendix B.

Through these experiments, we demonstrate that ABS is particularly effective for large-scale models. Specifically, ABS enables high-capacity RL agents to stably exploit large batch sizes—an advantage that has long been established

in supervised learning but has remained difficult to harness in RL due to the inherent non-stationarity of the learning process. By adaptively scheduling the rollout length in response to Behavioral Divergence, ABS bridges this gap, allowing scaled RL models to benefit from the same large-batch training advantages that have driven the success of large-scale supervised learning, including more stable gradient estimates, improved sample diversity, and stable convergence. Consequently, by unlocking stable access to large batch sizes, ABS fully realizes the potential of scaled RL: allowing high-capacity models to exploit their full representational capacity, and ultimately translating model scale into tangible performance gains.

### 5.5. *A3: Hyperparameter Sensitivity*

Figure 5 presents the hyperparameter sensitivity of ABS with respect to the rollout range $L$ (Top), policy change thresholds $\delta$ (Center), and adapt frequency $K$ (Bottom) on Atari-10.

**Rollout Range** $[L_{min}, L_{max}]$. As shown in Figure 5 (Top), setting $L_{min}$ too small (8; 1,024 samples) or large (32; 4,096 samples) leads to high performance variance across environments. With an appropriate $L_{min}$ of 16 (2,048 samples), setting $L_{max} = 64$ (8,192 samples) yields the best overall performance (see also Figure 4).

**Policy Change Thresholds** $[\delta_{min}, \delta_{max}]$. Figure 5 (Center) shows that (15%–85%) achieves the highest average performance, with (5%–95%) as a close second. However, (15%–85%) exhibits performance degradation on *BattleZone-v5*, whereas (5%–95%) yields consistent improvements across all environments. We therefore adopt (5%–95%) as the default setting for policy change thresholds.

**Adapt Frequency** $K$. Figure 5 (Bottom) shows that both excessively small and large values of $K$ lead to performance degradation in certain environments. We therefore select $K = 50$ as the default adapt frequency, as it achieves the most stable performance across all environments.

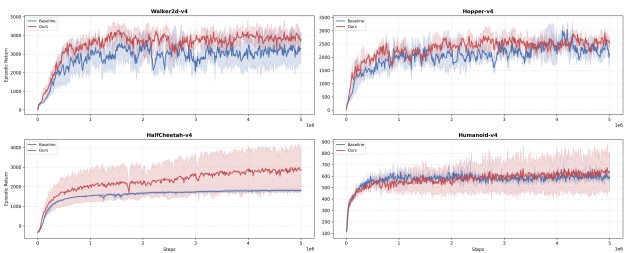

*Figure 6.* **Learning Curves of Vanilla PPO, with ABS** on four MuJoCo locomotion tasks. ABS (Ours) consistently enhances both the final performance and sample efficiency of PPO across all evaluated tasks. These results demonstrate that ABS can be generalized to continuous action spaces, beyond the discrete PQN setting.

### 5.6. A4: Application to Continuous Control (PPO)

To demonstrate the generality of our approach beyond discrete action spaces and value-based methods, we apply ABS to **PPO** on continuous control tasks (MuJoCo). In continuous spaces, measuring exact action mismatch is infeasible. Therefore, we revise our metric to use the **KL Divergence** between the old and new policies as a proxy for non-stationarity:

$$\delta_\pi \approx \mathbb{E}_{s \sim \mathcal{B}}[D_{\mathrm{KL}}(\pi_\theta(\cdot|s)||\pi_{\theta_{\mathrm{old}}}(\cdot|s))] \tag{7}$$

Figure 6 shows the learning curves for MuJoCo locomotion environments. PPO+ABS demonstrates stable learning and improved sample efficiency compared to standard PPO. This confirms that ABS is algorithm-agnostic and applicable to continuous domains. Further implementation details and specific hyperparameter settings are provided in Appendix A and Appendix C, respectively.

### 5.7. A4: Application to Replay Buffer-Style RL (BTR)

To demonstrate the generality of ABS beyond RL which does not use a replay buffer for policy update, we apply it to **BTR** (Clark et al., 2024) with minimal modification. For this replay buffer-style RL, the only architectural change is replacing the rollout range with a batch size range (*i.e.*, $[L_{\mathrm{min}}, L_{\mathrm{max}}] \rightarrow [B_{\mathrm{min}}, B_{\mathrm{max}}]$), leaving all other components unchanged. For detailed hyperparameter settings, please refer to Table 9.

Table 3 reports the IQM HNS with 95% confidence intervals on Atari-5 (Aitchison et al., 2023). Each configuration is trained over three independent seeds and evaluated for 100 episodes per seed. Despite the minimal adaptation required to apply ABS to an off-policy setting, ABS improves performance in 4 out of 5 environments, with the remaining environment showing comparable scores to the baseline. These results suggest that ABS effectively generalizes across various RL algorithms—both with and without replay buffers.

*Table 3.* **Performance Improvement on BTR**, off-policy RL, on Atari-5. BTR with our ABS outperforms in most environments.

| Environment | BTR | BTR + ABS(Ours) |
|---|---|---|
| BattleZone-v5 | $2.313 \pm 0.643$ | $\mathbf{2.656 \pm 0.756}$ |
| DoubleDunk-v5 | $7.855 \pm 0.809$ | $\mathbf{17.976 \pm 1.727}$ |
| NameThisGame-v5 | $3.893 \pm 0.229$ | $\mathbf{4.143 \pm 0.065}$ |
| Phoenix-v5 | $25.784 \pm 7.292$ | $\mathbf{29.293 \pm 13.279}$ |
| Qbert-v5 | $\mathbf{1.997 \pm 0.097}$ | $1.956 \pm 0.097$ |

## 6. Conclusion

In this paper, we proposed a novel training framework designed to training scaled Reinforcement Learning (RL) models. We identified the inherent non-stationarity of early training as the primary bottleneck preventing the adoption of large batch sizes in RL. By analyzing this phenomenon, we demonstrated that while small batches are essential for preserving model plasticity and minimizing bias during the volatile early stages, large batches become crucial for precise convergence as the policy stabilizes.

Our proposed method, Adaptive Batch Scaling (ABS), effectively navigates this trade-off by dynamically scaling the batch size based on the Behavioral Divergence—a proxy for non-stationarity. Empirical evaluations confirm that this approach not only accelerates convergence and improves final performance but also enables the stable training of high-capacity backbone models. We hope this work serves as a stepping stone toward scalable Reinforcement Learning, and opening new avenues for training high-capacity RL agents more effectively.

**Limitations & Future Work**   While this work focuses on some online RL algorithms, several promising directions remain for future research. First, we plan to extend our ABS, especially replay buffer-style, to offline RL settings. Second, although our experiments were conducted on MLP and CNN-based backbones, we aim to validate our approach on Transformer-based architectures. Ultimately, we envision applying this methodology to Large Language Models, integrating it with emerging alignment techniques such as GRPO to facilitate scalable and efficient RLHF training.

## Impact Statement

This work presents Adaptive Batch Scaling (ABS), which successfully unlocks large-batch training and model scaling in Reinforcement Learning (RL). This scalability paves the way for training larger, more capable RL agents for complex real-world applications such as robotics, autonomous driving, and industrial optimization. While accelerating the deployment of powerful RL systems highlights the need for continued vigilance regarding safety and alignment protocols in automated decision-making, the enhanced efficiency of ABS fundamentally contributes to more accessible and computationally sustainable AI research.

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

# A. Implementation Details

### A.1. PQN (+ABS)

To ensure the stability of the training process and prevent performance degradation due to sudden shifts in the data distribution, we incorporate three stabilizing mechanisms into the Adaptive Batch Scaling (ABS) framework.

**Smoothing the Policy Divergence Metric**    While the instantaneous behavioral divergence $\delta_\pi$ provides a signal for non-stationarity, it can be susceptible to high-frequency noise inherent in stochastic environments. To mitigate this, we utilize a sliding window average of the last $W = 10$ measured divergences:

$$\bar{\delta}_\pi = \frac{1}{W} \sum_{i=1}^{W} \delta_{\pi,i} \tag{8}$$

During the initial 'burn-in' phase of training (where fewer than $W$ samples of $\delta_\pi$ are available), we fix the rollout length to $L_{\min}$. This ensures that the agent has established a baseline behavioral pattern before the adaptive scaling mechanism takes effect.

**Exponential Moving Average for Rollout Transitions**    Even with a smoothed divergence metric, a direct application of $L_{adpt}$ can lead to sudden spikes in the batch size, which may destabilize the gradient updates. We apply an Exponential Moving Average (EMA) to the rollout length updates:

$$L_{\text{curr}} \leftarrow (1 - \beta)L_{\text{prev}} + \beta L_{\text{adapt}} \tag{9}$$

where $\beta \in (0, 1]$ is the smoothing coefficient. This ensures a gradual transition between different batch scales, preserving the numerical stability of the optimizer.

**Compensation for Sample Efficiency**    In on-policy RL, the number of gradient updates per sample is a critical factor for sample efficiency. Since we keep the number of mini-batches $N_{\text{mb}}$ constant, increasing the rollout length $L$ naturally leads to fewer updates per transition if the number of epochs remains fixed. To maintain a consistent 'gradient intensity' (updates per transition), we scale the number of update epochs $N_{\text{epochs}}$ proportionally to the rollout length. Specifically, when $L_{\text{adapt}}$ increases from its baseline, $N_{\text{epochs}}$ is adjusted as follows:

$$N_{\text{epochs}}^{\text{scaled}} = N_{\text{epochs}}^{\text{base}} \times \frac{L_{\text{adapt}}}{L_{\min}} \tag{10}$$

For example, a transition from $L = 32$ with $N_{\text{epochs}} = 2$ to $L = 64$ would result in $N_{\text{epochs}} = 4$. This compensation ensures that the model continues to extract sufficient information from the collected experience even as the batch size grows.

## A.2. PQN (+GNS)

---

**Algorithm 2** Dynamic Batch Size with GNS Estimation

---

**Input:** Current batch data $\mathcal{D}$, Current batch size $B$, Number of micro-batches $m$, Number of environments $E$ and Rollout range $[L_{\min}, L_{\max}]$

Set $B_{\min} = E \cdot L_{\min}$, $B_{\max} = E \cdot L_{\max}$

Split $\mathcal{D}$ into $m$ micro-batches $\{\mathcal{D}_1, \ldots, \mathcal{D}_m\}$ of size $b = B/m$

Compute gradients $g_i = \nabla \mathcal{L}(\theta; \mathcal{D}_i)$ for $i = 1 \ldots m$

Compute mean gradient $\bar{g} \leftarrow \frac{1}{m} \sum g_i$

Estimate Noise: $N \leftarrow b \cdot \frac{1}{m-1} \sum \|g_i - \bar{g}\|^2$

Estimate Signal: $S \leftarrow \|\bar{g}\|^2 - \frac{1}{B} N$

Calculate GNS: $\hat{\mathcal{B}}_{\text{simple}} \leftarrow N/S$

Update Batch Size: $B_{\text{new}} \leftarrow \text{clip}(\lfloor \hat{\mathcal{B}}_{\text{simple}} \rfloor, B_{\min}, B_{\max})$

---

We employ a dynamic batch size adjustment strategy based on the *Gradient Noise Scale* (GNS), denoted as $\mathcal{B}_{\text{simple}}$, following the empirical model proposed by McCandlish et al. (2018). The GNS serves as a critical metric for determining the optimal batch size where the noise in the stochastic gradient estimates begins to dominate the useful signal. It is defined as the ratio of the trace of the gradient covariance matrix $\Sigma$ to the squared norm of the true gradient $G$:

$$\mathcal{B}_{\text{simple}} = \frac{\text{tr}(\Sigma)}{\|G\|^2} \tag{11}$$

where $G = \mathbb{E}[\nabla \mathcal{L}(\theta)]$ represents the true gradient over the data distribution, and $\Sigma = \text{Cov}(\nabla \mathcal{L}(\theta))$ is the covariance matrix of the per-sample gradients.

**Efficient Estimation Strategy** Direct computation of $\Sigma$ and $G$ is computationally prohibitive. To estimate $\mathcal{B}_{\text{simple}}$ efficiently during training, we utilize a variance-based estimator using micro-batches. At every update step (or at a fixed frequency), the current batch $B$ is split into $m$ micro-batches, each of size $b = B/m$. Let $g_i$ denote the gradient computed from the $i$-th micro-batch.

We first compute the mean gradient of the micro-batches, $\bar{g} = \frac{1}{m} \sum_{i=1}^{m} g_i$. The noise term, $\text{tr}(\Sigma)$, is estimated using the sample variance of the micro-batch gradients, scaled by the micro-batch size:

$$\text{tr}(\Sigma) \approx b \cdot \frac{1}{m-1} \sum_{i=1}^{m} \|g_i - \bar{g}\|^2 \tag{12}$$

The signal term, $\|G\|^2$, is estimated by correcting the bias in the squared norm of the mean gradient:

$$\|G\|^2 \approx \|\bar{g}\|^2 - \frac{1}{B} \text{tr}(\Sigma) \tag{13}$$

Finally, the estimated GNS is given by:

$$\hat{\mathcal{B}}_{\text{simple}} = \frac{b \sum_{i=1}^{m} \|g_i - \bar{g}\|^2}{(m-1)\|\bar{g}\|^2 - \frac{b}{B} \sum_{i=1}^{m} \|g_i - \bar{g}\|^2} \tag{14}$$

The complete procedure is summarized in Algorithm 2.

## A.3. PPO (+ABS)

---

**Algorithm 3** PPO with Adaptive Rollout Scheduling (ARS)

---

**Input:** Initial parameters $\theta$, environments $E$, update frequency $K$, KL thresholds $\delta_{\min,\text{KL}}, \delta_{\max,\text{KL}}$, rollout bounds $L_{\min}, L_{\max}$.

Initialize rollout length $L_{\text{adapt}} = L_{\min}$, global step $t = 0$.

**while** $t < T_{\text{total}}$ **do**
   If $t \pmod{K} == 0$, set $\theta_{\text{old}} \leftarrow \theta$.

   **Step 1: Data Collection**
   Collect trajectories $\mathcal{D}$ for $L_{\text{adapt}}$ steps using $E$ environments and policy $\pi_\theta$.
   $t \leftarrow t + (E \times L_{\text{adapt}})$

   **Step 2: Policy Update (PPO)**
   **for** $epoch = 1$ **to** $N_{\text{epochs}}$ **do**
      Sample mini-batches from $\mathcal{D}$.
      Update $\theta$ by maximizing PPO objective $\mathcal{L}^{\text{CLIP}}$.
   **end for**

   **Step 3: Adaptive Scaling**
   **if** $t \pmod{K} == 0$ **then**
      Calculate KL Divergence between current and old policy:
         $\delta_{\pi,\text{KL}} = \mathbb{E}_{s\sim\mathcal{D}}[D_{KL}(\pi_\theta(s)||\pi_{\theta_{\text{old}}}(s))]$

      Determine target length $L_{\text{target}}$ using log-linear interpolation:
         Calculate $L'_{\text{adapt}}$ based on $\delta_{\min,\text{KL}}, \delta_{\max,\text{KL}}$ and $\delta_{\pi,\text{KL}}$ (Eq. 4).

      Smooth update: $L_{\text{adapt}} \leftarrow (1-\alpha)L_{\text{adapt}} + \alpha L'_{\text{adapt}}$
   **end if**
**end while**

---

While both PQN and PPO benefit from dynamic horizon adjustments, the distinct nature of their action spaces—discrete for PQN and continuous for PPO—necessitates different metrics for measuring Behavioral Divergence ($\delta$). Here, we contrast the *Adaptive Batch Scaling (ABS)* used in PQN with the *Adaptive Rollout Scaling (ARS)* proposed for PPO.

**Metric Selection: Behavioral vs. KL Divergence** For PQN, which operates in a discrete action space with a deterministic Q-policy, we employed *Behavioral Divergence* as the stability metric. This measures the fraction of states in a reference batch where the greedy action changes between updates: $\delta_\pi = \mathbb{E}[\mathbb{I}(\pi_\theta(s) \neq \pi_{\theta_{\text{old}}}(s))]$. This binary metric is computationally efficient and directly captures the instability of discrete policies.

In contrast, PPO operates in a continuous action space where the policy $\pi_\theta(\cdot|s)$ is typically modeled as a Gaussian distribution $\mathcal{N}(\mu_\theta(s), \sigma_\theta(s))$. A binary comparison of actions is unsuitable here. Instead, we utilize the *Kullback-Leibler (KL) Divergence* to measure the distributional shift between the current and the previous policy snapshot. The metric $\delta_{\text{PPO}}$ is computed analytically as:

$$\delta_{\pi,\text{KL}} = \mathbb{E}_{s\sim\mathcal{D}}\left[D_{KL}(\pi_\theta(\cdot|s) \,||\, \pi_{\theta_{\text{old}}}(\cdot|s))\right] \tag{15}$$

# B. Additional Results

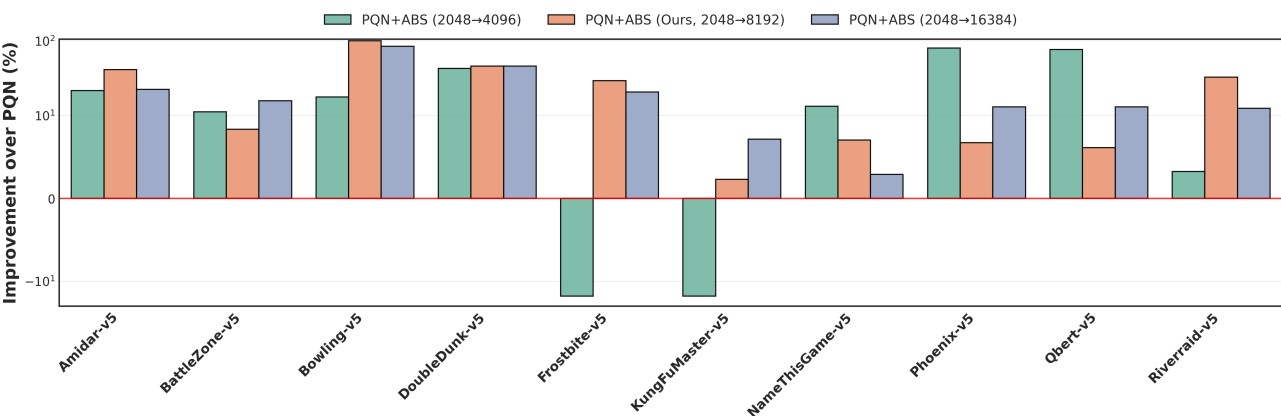

*Figure 7.* Log Scaled Performance Improvement of PQN with ABS over various $L_{\max}$ relative to vanilla PQN on the Atari-10 benchmark.

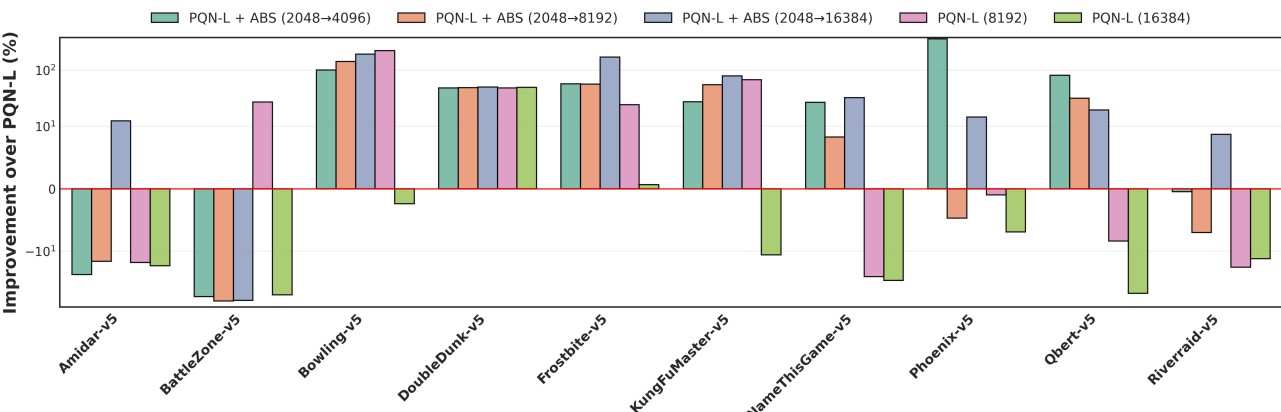

*Figure 8.* Log Scaled Performance Improvement of PQN-L with large fixed batches and ABS over various $L_{\max}$ relative to vanilla PQN-L on the Atari-10 benchmark.

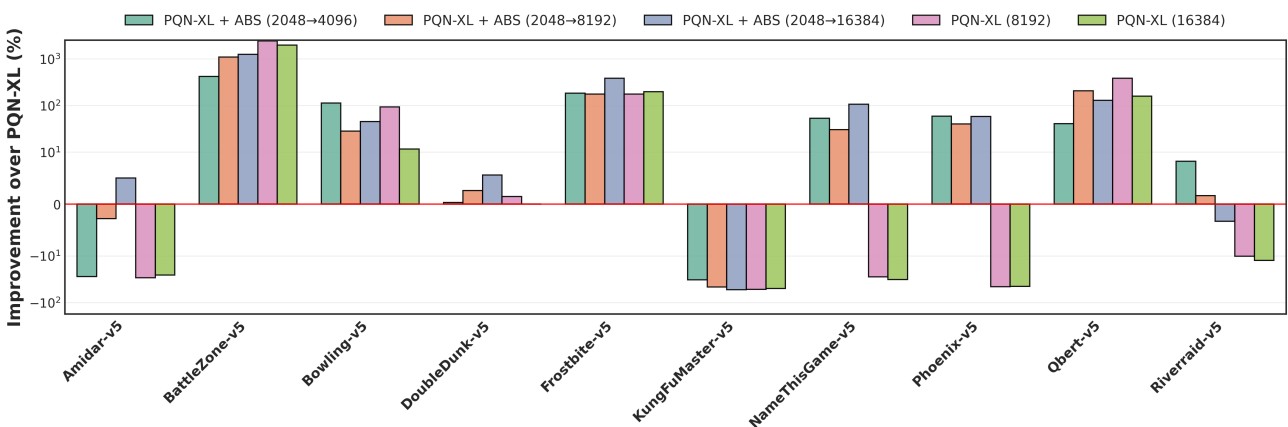

*Figure 9.* Log Scaled Performance Improvement of PQN-XL with large fixed batches and ABS over various $L_{\max}$ relative to vanilla PQN-XL on the Atari-10 benchmark.

*Table 4.* Early (first 10%) and Late (last 10%) training performance across 10 environments.

| Environment | Baseline | | | | Ours | | | |
|---|---|---|---|---|---|---|---|---|
| | Early 10% Mean ↑ | Early 10% Std ↓ | Late 10% Mean ↑ | Late 10% Std ↓ | Early 10% Mean ↑ | Early 10% Std ↓ | Late 10% Mean ↑ | Late 10% Std ↓ |
| Amidar | 24.59 | 0.82 | 568.73 | 105.26 | **27.87** | **0.53** | **695.08** | **79.69** |
| BattleZone | **2,674.57** | 111.91 | 28,275.76 | **639.56** | 2,607.77 | 106.43 | **29,932.65** | 1,016.06 |
| Bowling | **26.01** | 0.75 | **14.74** | 20.45 | 25.46 | **0.54** | 22.02 | **20.22** |
| DoubleDunk | **-18.17** | 0.14 | **-37.63** | 8.11 | -18.20 | 0.14 | -21.98 | **0.77** |
| Frostbite | 145.07 | **1.64** | 2,430.97 | 446.38 | **158.63** | 4.73 | **2,947.01** | **342.58** |
| KungFuMaster | 2,688.73 | 99.47 | **37,651.88** | 5,484.19 | **2,909.41** | **73.72** | 33,027.95 | **1,154.99** |
| NameThisGame | 2,612.14 | **39.90** | 12,857.59 | **165.24** | **2,666.56** | 51.69 | **13,554.39** | 1,195.49 |
| Phoenix | 1,252.02 | 44.25 | 5,181.59 | **204.33** | **1,529.88** | **12.69** | **5,544.88** | 355.20 |
| Qbert | 308.49 | 15.42 | 4,389.09 | 174.82 | **333.30** | **6.77** | **4,614.49** | **110.13** |
| Riverraid | 1,912.26 | **1.69** | 8,779.16 | **556.13** | **2,002.44** | 8.89 | **11,052.94** | 1,166.14 |

## C. Hyperparameters

Tables 5 through 8 provide the detailed hyperparameter configurations used for our experiments on the Atari benchmark. Tables 5 and 6 list the settings for the standard PQN architecture and the high-capacity Multi-Skip Residual MLP backbone, respectively. Tables 7 specifies the hyperparameters for the PQN, incorporating GNS based dynamic batch scheduling method. Additionally, Table 8 specifies the hyperparameters for the PPO baseline, incorporating our proposed adaptive rollout mechanism to evaluate its scalability.

*Table 5.* PQN (+ Ours) Hyperparameters

| Parameter | Value |
|---|---|
| *General Hyperparameters* | |
| Total Timesteps | $2 \times 10^7$ |
| Num. Environments | 128 |
| Frame Stack | 4 |
| Sticky Action Probability | 0 |
| Life Information | False |
| Learning Rate | $2.5 \times 10^{-4}$ |
| Base Steps per Rollout ($L_{\min}$) | 32 |
| Discount Factor ($\gamma$) | 0.99 |
| Mini-batches | 4 |
| Update Epochs ($N_{epochs}$) | 2 |
| Max Grad Norm | 10.0 |
| Exploration ($\epsilon_{\text{start}} \rightarrow \epsilon_{\text{end}}$) | $1.0 \rightarrow 0.001$ |
| Exploration Fraction | 0.10 |
| Q-Learning $\lambda$ | 0.65 |
| *Adaptive Rollout Hyperparameters* | |
| Adapt Rollout | True |
| Rollout Range ($L_{\min}, L_{\max}$) | $[16, 64]$ |
| Adapt Frequency ($K$) | 50 iterations |
| Policy Change Thresholds ($\delta_{\min}, \delta_{\max}$) | $[0.05, 0.95]$ |
| Schedule Type | log |

*Table 6.* PQN (+ Multi-Skip + Ours) Hyperparameters

| Parameter | Value |
|---|---|
| *General Hyperparameters* | |
| Total Timesteps | $2 \times 10^7$ |
| Num. Environments | 128 |
| Frame Stack | 4 |
| Sticky Action Probability | 0 |
| Life Information | False |
| Learning Rate | $2.5 \times 10^{-4}$ |
| Anneal LR | False |
| Base Steps per Rollout ($L_{\min}$) | 32 |
| Discount Factor ($\gamma$) | 0.99 |
| Mini-batches | 4 |
| Update Epochs ($N_{epochs}$) | 2 |
| Max Grad Norm | 10.0 |
| Exploration ($\epsilon_{\text{start}} \to \epsilon_{\text{end}}$) | $1.0 \to 0.001$ |
| Exploration Fraction | 0.10 |
| Q-Learning $\lambda$ | 0.65 |
| *Multi-Skip Network Architecture* | |
| Use Multi-Skip Residual MLP | True |
| MLP Hidden Size | 512 |
| MLP Layers | 5 |
| Layer Normalization | True |
| Activation Function | ReLU |
| CNN Channels | $(64, 128, 128)$ |
| *Adaptive Rollout Settings* | |
| Adapt Rollout | True |
| Rollout Range ($L_{\min}, L_{\max}$) | $[16, 128]$ |
| Adapt Frequency ($K$) | 50 iterations |
| Policy Change Thresholds ($\delta_{\min}, \delta_{\max}$) | $[0.05, 0.95]$ |
| Schedule Type | log |

*Table 7.* PQN (+ GNS) Hyperparameters

| Parameter | Value |
|---|---|
| *General Hyperparameters* | |
| Total Timesteps | $2 \times 10^7$ |
| Learning Rate | $2.5 \times 10^{-4}$ |
| Num. Environments | 128 |
| Base Steps per Rollout ($L_{\min}$) | 32 |
| Discount Factor ($\gamma$) | 0.99 |
| Mini-batches | 4 |
| Update Epochs ($N_{epochs}$) | 2 |
| Max Grad Norm | 10.0 |
| Exploration ($\epsilon_{\text{start}} \to \epsilon_{\text{end}}$) | $1.0 \to 0.001$ |
| Exploration Fraction | 0.10 |
| Q-Learning $\lambda$ | 0.65 |
| *GNS Hyperparameters* | |
| GNS | True |
| Rollout Range ($L_{\min}, L_{\max}$) | $[16, 64]$ |
| Adapt Frequency ($K$) | 50 iterations |

*Table 8.* PPO (+ Ours) Hyperparameters

| Parameter | Value |
|---|---|
| *General PPO Hyperparameters* | |
| Total Timesteps | $5 \times 10^6$ |
| Learning Rate ($\alpha$) | $3 \times 10^{-4}$ |
| Num. Environments | 1 |
| Base Steps per Rollout | 2048 |
| Anneal LR | True |
| Discount Factor ($\gamma$) | 0.99 |
| GAE Lambda ($\lambda$) | 0.95 |
| Mini-batches | 32 |
| Update Epochs ($K$) | 10 |
| Advantage Normalization | True |
| Clip Coefficient ($\epsilon$) | 0.2 |
| Clip Value Loss | True |
| Entropy Coef. ($c_2$) | 0.0 |
| Value Function Coef. ($c_1$) | 0.5 |
| Max Grad Norm | 0.5 |
| Target KL | None |
| Eval Episodes | 10 |
| *Adaptive Rollout Settings* | |
| Adapt Rollout | True |
| Rollout Range ($L_{\min}, L_{\max}$) | $[1024, 8192]$ |
| Adapt Frequency (K) | 10 iterations |
| KL Thresholds ($D_{\text{KL}}^{\text{low}}, D_{\text{KL}}^{\text{high}}$) | $[0.01, 0.1]$ |

*Table 9.* BTR (+ Ours) Hyperparameters

| Parameter | Value |
|---|---|
| *General BTR Hyperparameters* | |
| Total Timesteps | $2 \times 10^7$ |
| Num. Environments | 64 |
| Frame Stack | 4 |
| Sticky Action Probability | 0.25 |
| Life Information | False |
| Batch Size | 256 |
| Replay Ratio | 1 |
| Learning Rate ($\alpha$) | $1 \times 10^{-4}$ |
| Discount Factor ($\gamma$) | 0.997 |
| N-step Returns | 3 |
| Max Grad Norm | 10.0 |
| *Network Architecture* | |
| Backbone | IMPALA |
| Model Size (width multiplier) | 2 |
| Linear Layer Size | 512 |
| Max Pooling | True |
| Max Pool Size | 6 |
| Activation | ReLU |
| Noisy Layers | True |
| Spectral Normalization | True |
| Dueling Network | True |
| *Algorithm Components* | |
| Distributional RL | IQN |
| Num. Quantile Samples ($N$) | 8 |
| Num. Cosine Features | 64 |
| Munchausen RL | True |
| Munchausen $\alpha$ | 0.9 |
| Double Q-Learning | False |
| *Prioritized Experience Replay (PER)* | |
| PER | True |
| Priority Exponent ($\alpha_{\text{PER}}$) | 0.2 |
| Beta Annealing | False |
| *Target Network* | |
| Target Update Strategy | Hard Update |
| Target Update Frequency | 500 gradient steps |
| *Adaptive Batch Settings* | |
| Adapt Batch | True |
| Batch Size Range ($B_{\min}, B_{\max}$) | $[64, 1024]$ |
| Adapt Frequency (K) | 50 iterations |
| Policy Change Thresholds ($\delta_{\min}, \delta_{\max}$) | $[0.05, 0.95]$ |

