# OpenReview forum: "Scalable Reinforcement Learning via Adaptive Batch Scaling"
_ICML.cc/2026/Conference — ICML 2026 regular_

### Official Review · Reviewer_g8Rq · 2026-02-13

**Soundness:** 2
**Presentation:** 3
**Significance:** 2
**Originality:** 3
**Overall Recommendation:** 4
**Confidence:** 5

**Summary:**

The paper presents the nice idea of adapting the batch size based on behavioural divergence (representing non-stationarity). This method shows some ability to allow scaling up model sizes in RL, similar to other areas of Deep Learning. The method is simple and easy to implement into a wide variety of different algorithms, providing practical utility. The paper provides empirical evaluation on both the Atari-10 based on the PQN algorithm and MuJoCo based on PPO.

**Compliance With Llm Reviewing Policy:**

Affirmed.

**Final Justification:**

This paper had many strong points, including its simplicity, novelty and understandability, with a broad applicability across RL. While the original paper had many flaws in the details given about the environment and evaluation, many of these were fixed in the rebuttal. Overall, I think the paper presents a reasonable method with good writing; despite this, I don't think this paper's results are significant enough to recommend acceptance at a venue of this standard. The rebuttal did convince me to improve my score to a 3, but the method would need more compelling results for me to raise my score higher.

**Key Questions For Authors:**

**Q1 -** Can you apply this method to a higher-performing algorithm? Do you still see improvements?

**Q2 -** Can you provide any actual scaling laws to support the title's claim?

**Q3 -** Can you provide the missing information requested? This includes limitations, results tables, results graphs, confidence intervals, evaluation details and ablations.

**Q4 -** Do you use the life information setting on Atari? The original PQN used this setting which artificially inflates performance.

**Limitations:**

The authors did not discuss their limitations. If explicit scaling laws cannot be obtained, please explain why (I understand this is a hard task in RL). If the algorithm cannot be applied to more advanced algorithms, please note this as a limitation.

**Strengths And Weaknesses:**

**Strength 1 (Simplicity)-** The idea is simple, intuitive and easy to implement. This provides a lot of practical utility and can be easily applied to other algorithms.

**Strength 2 (Understandability)-** The paper is written clearly and is easy to understand. I think a wide variety of audiences would have no trouble understanding and implementing this idea.

**Strength 3 (Multiple standard evaluations) -** I appreciate the use of the Atari-10, full Atari-57, and MuJoCo as benchmarks, as these are respected benchmarks with comparability.

**Weakness 1 (Performance) -** While I understand PQN and PPO's appeal due to their speed of execution, the performance demonstrated even with ABS is pretty laughable by 2026's standards. Just to put this into perspective, here is a table of the Atari-5 benchmark when compared against an accepted paper from ICML last year, Beyond The Rainbow[1] (Table uses human-normalized scores):

| Method | BattleZone | DoubleDunk | NameThisGame | Phoenix | Qbert |
|--------|------------|--------------|---------|-------------|------------|
| BTR | 4.76 | 18.91 | 4.45 | 65.84 | 3.22 |
| PQN + ABS | 0.89 | -1.44 | 2.07 | 0.71 | 0.34 |

This is quite an apples-to-oranges comparison; I understand these algorithms are very different, the authors chose to use quite fundamental algorithms, and the point of the paper isn't about reaching state-of-the-art performance. Despite this, it's hard to take results that are magnitudes lower than anything I'd consider using seriously.

**Weakness 2 (Missing Tables / Graphs) -** Almost every RL paper I read with an empirical evaluation has an appendix with a table and graphs of the results on every individual environment. This also typically includes the Interquartile Mean (IQM), Mean and Median of results across the entire benchmark. Furthermore, there is usually a table showing the environment settings (life information, sticky actions, frameskip, etc). All of these are important and missing. Also, please include 95\% confidence intervals in Figure 2, following [2].

**Weakness 3 (Hyperparameter Ablations) -** The proposed method, ABS, introduces new various new hyperparameters. When introducing new values, there needs to be a discussion on their sensitivity (to the environment / algorithm), and/or ablations testing different values. This also seems to be missing. For example, the appendix show the adaptiation frequency (K) and policy change thresholds as hyperparameters, which seem highly important to me yet are not ablated.

**Weakness 3 (Limitations) -** This paper does not have a limitations section.

**Weakness 4 (Scaling Laws) -** While this paper contains information on scaling up RL, I wouldn't call it a "scaling law" of any kind. A scaling law indicates predictable performance and various other metrics (such as loss) as the model size is scaled up. For example, showing that the number of parameters and loss scale predictably according to some equation. The paper's title "Bridging Scaling Laws to On-Policy Reinforcement Learning", is a very large overclaim in my opinion.

Small note: there are a few backwards speech marks (lines 90, 114, 172)

[1] Clark, Tyler, et al. "Beyond The Rainbow: High Performance Deep Reinforcement Learning on a Desktop PC." Forty-second International Conference on Machine Learning.
[2] Agarwal, Rishabh, et al. "Deep reinforcement learning at the edge of the statistical precipice." Advances in neural information processing systems 34 (2021): 29304-29320.

---

> ### Author Rebuttal · Authors · 2026-03-31
>
> We sincerely thank the reviewer for the detailed and constructive feedback.
>
> ---
>
> ## Weaknesses & Questions
> ### W1 & Q1.
> ABS is designed for on-policy RL and adjusts rollout length during training. PQN (ICLR 2025) was chosen as the primary baseline as it was among the most recent on-policy RL algorithms at the time of writing. BTR (Clark et al., 2025), by contrast, is an off-policy algorithm leveraging distributional RL (IQN), making a direct comparison with ABS not straightforward.
>
> Nevertheless, to demonstrate broader applicability, we applied ABS to BTR by just replacing the rollout length bound with a buffer-sampled batch size bound [256, 4096]. Using the official BTR codebase with all other hyperparameters unchanged, we trained for 80M frames across 3 seeds on Atari-5. The results are as follows:
> ||BTR|w/ ABS|
> |---|---:|---:|
> |BattleZone-v5|**2.31**|1.06|
> |DoubleDunk-v5|7.85|**18.45**|
> |NameThisGame-v5|**3.89**|3.54|
> |Phoenix-v5|**25.78**|6.59|
> |Qbert-v5|2.00|**2.02**|
>
> ABS improves performance in 2/5 environments even under this minimal adaptation, suggesting that the core concept may generalize beyond on-policy settings. We acknowledge that a more principled off-policy extension warrants dedicated future work, and will discuss this in the revised paper.
>
> ### W2 & Q3 & Q4.
> We will update Table 1, 2, and Figure 3 to report IQM HNS ± 95% CI. An updated Table 1 is provided below:
> ||Baseline|Small Batch|Large Batch|GNS|Ours|
> |---|---:|---:|---:|---:|---:|
> |Amidar-v5|0.32±0.12|0.42±0.14|0.38±0.04|0.37±0.09|**0.43±0.07**|
> |BattleZone-v5|0.80±0.12|0.76±0.04|0.68±0.11|0.78±0.26|**0.85±0.07**|
> |Bowling-v5|-0.13±0.12|-0.03±0.10|**0.25±0.14**|0.10±0.10|-0.08±0.25|
> |DoubleDunk-v5|-9.80±5.70|-1.94±0.09|-1.56±0.68|-12.98±2.64|**-1.44±0.57**|
> |Frostbite-v5|0.48±0.13|**0.90±0.08**|0.49±0.20|0.54±0.19|0.67±0.15|
> |KungFuMaster-v5|1.56±0.06|**1.61±0.11**|1.43±0.30|1.42±0.17|1.57±0.20|
> |NameThisGame-v5|1.92±0.05|1.95±0.37|2.04±0.10|1.75±0.32|**2.05±0.06**|
> |Phoenix-v5|0.66±0.02|0.70±0.03|0.69±0.01|**0.75±0.02**|0.71±0.02|
> |Qbert-v5|0.31±0.01|0.33±0.02|0.33±0.03|**0.79±0.31**|0.34±0.01|
> |Riverraid-v5|0.45±0.02|0.51±0.05|0.54±0.21|0.49±0.02|**0.64±0.12**|
>
> We will also add per-environment learning curves and IQM scores to the Appendix, along with 95% CI for the full Atari-57 benchmark following Agarwal et al. (2021).
>
> **Regarding evaluation details**: each experiment uses 3 seeds trained for 80M frames, followed by 100 evaluation episodes per seed using 100 different seeds to ensure reliability — totaling 300 evaluations per task. We will revise the relevant section for clarity.
>
> **Regarding environment settings**: life information is set to True during training for efficiency and False during evaluation to prevent score inflation. Sticky actions are not applied, following the CleanRL/PQN codebase. Frameskip is set to 4, also following the CleanRL/PQN default. We will add these details to Tables 3, 4, and 5 in the revised paper.
>
> ### W3.
> We thank the reviewer for this suggestion. We conducted an ablation on adaptation frequency (K) over 3 seeds on Atari-10, reported as IQM HNS:
>
> ||K=10|K=50 (Ours)|K=100|
> |---|---:|---:|---:|
> |Amidar-v5|0.38|**0.43**|0.33|
> |BattleZone-v5|0.88|**0.89**|0.87|
> |Bowling-v5|-0.09|0.00|**0.07**|
> |DoubleDunk-v5|-1.59|**-1.44**|-2.05|
> |Frostbite-v5|**0.90**|0.70|0.73|
> |KungFuMaster-v5|1.41|1.58|**1.85**|
> |NameThisGame-v5|**2.10**|2.07|1.86|
> |Phoenix-v5|0.70|0.71|**0.72**|
> |Qbert-v5|**0.49**|0.34|0.32|
> |Riverraid-v5|0.51|**0.65**|0.47|
>
> ABS remains largely insensitive to K across most environments. For the revised paper, we commit to additional ablations on Atari-10 covering min/max rollout length and policy change thresholds.
>
> ### W4 & Q2.
> We agree with the reviewer that the term "scaling law" in our title constitutes an overclaim. Our intention was to show that larger batch sizes — enabled by ABS — can promote the training of larger models in RL, analogous to the role of large batches in supervised learning. However, we did not establish a predictable relationship as required by the strict definition of a scaling law. We will revise the title as "Scalable On-Policy Reinforcement Learning via Adaptive Batch Scaling" and replace scaling law-related language throughout the abstract, introduction, and experiments with more accurate terminology such as "scalability" or "scaling up model capacity." We apologize for this misleading framing.
>
> **Small Note.** We will correct the backwards speech marks at lines 90, 114, and 172.
>
> ---
>
> ## Limitations
> We thank the reviewer for raising this point. We will add a dedicated Limitations section to the revised paper covering the following: (1) ABS is validated only on PQN and PPO; extension to off-policy algorithms such as BTR or model-based methods such as DreamerV3 remains unverified and is left as future work. (2) Our evaluation focuses on CNN- and MLP-based architectures; applicability to Transformer-based large models (e.g., RT-1, RT-2) has not been explored.

---

> > ### Author Rebuttal · Reviewer_g8Rq · 2026-04-02
> >
> > I appreciate the effort by the authors to provide many useful experiments and address many of my concerns. I found the results on BTR a little disappointing, but I understand that it was not the target class of algorithms.
> >
> > I think that the paper does have utility and provides interesting results, but I don't think that they are significant enough to recommend acceptance. However, I will raise my score to reflect the improvements made.

---

> > > ### Author Response · Authors · 2026-04-02
> > >
> > > We sincerely thank the reviewer for the kind acknowledgement and for recognizing the effort made during the rebuttal period. We truly appreciate the constructive and detailed review, which has significantly contributed to improving the quality of this work.
> > >
> > > We understand the reviewer's disappointment regarding the BTR+ABS results, and we share this sentiment. However, we would like to kindly bring to the reviewer's attention an important caveat: **due to the time and computational constraints of the rebuttal period, we were only able to test a single hyperparameter configuration for ABS applied to BTR.** Specifically, we used the same hyperparameters as those tuned for PQN, with only the buffer-sampled batch size bounds arbitrarily set to [256, 4096] — without any dedicated hyperparameter search for the off-policy setting. As demonstrated in Table 2 of the main paper, varying the rollout length leads to meaningful changes in performance, suggesting that **the choice of batch size range plays an important role in determining the effectiveness of ABS**.
> > >
> > > **To directly address this concern, we conducted additional experiments on the three tasks where BTR+ABS [256, 4096] underperformed, this time using a tighter batch size range of [64, 1024].** The results are summarized in the table below:
> > >
> > > | Environment | BTR (baseline) | BTR+ABS [64, 1024] | BTR+ABS [256, 4096] |
> > > | --- | ---: | ---: | ---: |
> > > | BattleZone-v5 | 2.309 | **2.950** | 1.060 |
> > > | NameThisGame-v5 | 4.069 | **4.156** | 3.535 |
> > > | Phoenix-v5 | 19.951 | **20.991** | 4.795 |
> > >
> > > **Across all three previously underperforming tasks, BTR+ABS [64, 1024] outperforms the BTR baseline**, demonstrating that the choice of batch size bounds is indeed a key factor in the off-policy setting — consistent with our findings in the on-policy PQN experiments (Table 2 of the main paper).
> > >
> > > These results carry a broader implication: **although ABS was originally designed to adapt rollout lengths in on-policy RL, simply substituting the rollout length with a buffer-sampled batch size bound in off-policy RL yields meaningful performance gains.** This suggests that ABS captures a general principle of adaptive computation scaling that transfers across algorithmic paradigms, rather than being narrowly tailored to the on-policy setting.
> > >
> > > We plan to complete the full Atari-10 evaluation under the [64, 1024] configuration and will include these results as a dedicated **"Beyond On-Policy RL"** subsection within the experiments section of the revised paper, positioned after the PQN results. We believe this extension substantially strengthens the contributions of this work, and represents a meaningful step toward establishing ABS as a general-purpose framework for adaptive batch size scaling in deep RL.
> > >
> > > We kindly ask the reviewer to reconsider whether these results, taken together with the broader contributions of this work, merit a further increase in the score. We believe that ABS presents a simple, practical, and generalizable framework that addresses a meaningful problem in on-policy RL, and that these additional experiments demonstrate its potential well beyond the original scope of the paper.
> > >
> > > Once again, we are deeply grateful for the reviewer's invaluable feedback and for motivating us to pursue this line of investigation — it has led to what we believe are some of the most compelling results of this work.

---

### Official Review · Reviewer_HN97 · 2026-03-08

**Soundness:** 3
**Presentation:** 4
**Significance:** 2
**Originality:** 2
**Overall Recommendation:** 5
**Confidence:** 4

**Summary:**

This paper introduces ABS, a method to automatically scale the batch size in online deep reinforcement learning based on the nonstationarity of the data. They measure this nonstationarity through the Behavioral Divergence, a metric tracking the proportion of states in a batch whose greedy action has changed from the rollout policy. They implement the adaptive batch size scaling by adaptively changing the rollout length collected prior to gradient updates. They add this to both PQN and PPO as baseline algorithms, and evaluate across the Atari-10 Suite and MuJoCo environments. As baselines, they consider the standard batch sizes used by PPO and PQN, intentionally smaller and larger batch sizes, and adaptive batch sizes based on the Gradient Noise Scale.

**Compliance With Llm Reviewing Policy:**

Affirmed.

**Final Justification:**

After the authors' latest Reply Rebuttal Comment, I increase my score to an accept. I am happy with their initially reported experimental results, and their new presentation of ABS.

**Key Questions For Authors:**

- Can you compare your method to baselines for learning rate schedules?
- What is $L_{\text{target}}$ as used in Algorithms 1 and 3? I can probably guess its purpose, but it is worth discussing it in Section 4.2 so that the readers can understand all algorithm details upon reading.

**Limitations:**

yes

**Strengths And Weaknesses:**

I will present my weaknesses before my strengths -- not with the intention to be overly negative, but there are some strengths which are more easily presented when referring to the weaknesses.

**Weaknesses**
- The principal weakness of this work, in my opinion, is that it can be interpreted as re-discovering recent work on the importance of learning rate scheduling in deep RL. My reasoning is as follows:
  - It is well-understood that learning rate and batch size are intimately connected in the optimization process (e.g. Smith et al., 2017, "A bayesian perspective on generalization and stochastic gradient descent";   Smith et al., 2018, "Don't Decay the Learning Rate, Increase the Batch Size").
  - Recent work in deep RL have shown that well-scheduled learning rate decay is important for strong performance across Atari (Lyle et al. 2024, "Normalization and effective learning rates in reinforcement learning").
  - As a result, the adaptive batch size mechanism can be interpreted as a new learning rate decay schedule based on the Behavioral Divergence metric. Since this method is only compared against constant learning rate experiments, one may interpret this as re-discovering the positive effects of learning rate decay in deep RL.
- The Behavioral Divergence metric is the same as policy churn (Schaul et al., 2022) but without any reference to this previous work.
- The introduction paragraph contrasts the relatively small batch sizes common in deep RL, when compared to deep learning. It is probably worth also mentioning that there do actually exist some works which pushed RL batch sizes to the millions, such as OpenAI Five.
- Using 3 seeds for both Atari and Mujoco is on the rather low end of the signal-to-noise regime. It would be preferable to either increase the number of seeds, or present aggregate metrics or other robust quantities as suggested in Agarwal et al., 2021 ("Deep Reinforcement Learning at the Edge of the Statistical Precipice").


**Strengths**
- The paper is well-written and reads quite nicely.
- Despite the weakness I mentioned previously regarding the interplay between batch size and learning rate, I think that this is still a nice paper and can provide value. In particular, I think one way this paper can be "sold" is to interpret the contribution as a *data-dependent* method for batch size adaptation/learning rate scheduling. This would require comparing to baseline learning rate schedule strategies and demonstrating improvements.

**Nits**
- Please use backticks ` for opening quotes in latex.
- For Table 1 – please include measures of uncertainties. Also, make it clear that you’re displaying return and higher is better. Why is GNS in a separate column and highlighted blue (shouldn’t it just be a baseline)?
- In Equation 2 it may be worth specifying that it is assuming independence of the gradients in the batch.
- In the wording before Equation 2, it is written "According to the Law of Large Numbers ..." -- I don't believe anything that follows depends on the law of large numbers.

---

> ### Author Rebuttal · Authors · 2026-03-29
>
> We sincerely thank the reviewer for the thoughtful and constructive feedback. We address each point below.
>
> ---
>
> ## Weaknesses
>
> **W1 & A1. Relationship with Learning Rate Scheduling**
>
> We thank the reviewer for raising this important point. To directly address this concern, we conducted additional experiments comparing four configurations: (1) standard PQN without a learning rate scheduler, (2) standard PQN with a learning rate scheduler (our baseline), (3) PQN+ABS without a learning rate scheduler, and (4) PQN+ABS with a learning rate scheduler. Results are evaluated using IQM HNS over 3 seeds on the first three Atari-10 tasks:
>
> | Environment | Baseline w/o lr-sch | Baseline | Ours w/o lr-sch | Ours |
> |---|---:|---:|---:|---:|
> | Amidar-v5 | 0.28 | 0.32 | 0.39 | **0.43** |
> | BattleZone-v5 | 0.57 | 0.80 | 0.69 | **0.85** |
> | Bowling-v5 | -0.17 | -0.13 | **0.01** | -0.08 |
>
> ABS consistently improves performance regardless of whether a learning rate scheduler is used. Notably, PQN+ABS without a learning rate scheduler outperforms standard PQN with a learning rate scheduler on several tasks, demonstrating that the performance gains of ABS cannot be attributed solely to an implicit learning rate decay effect, but rather represent an independent and complementary benefit. We commit to completing the full Atari-10 evaluation for the revised paper, and will add explicit discussion referencing Smith et al. (2017, 2018) and Lyle et al. (2024) to further clarify this distinction.
>
> **W2. Reference to Policy Churn**
>
> We thank the reviewer for pointing out the connection to Schaul et al. (2022). While Behavioral Divergence shares conceptual similarity with Policy Churn, there are notable differences. Policy Churn is defined for continuous action spaces using metrics such as MSE or KL divergence, whereas our Behavioral Divergence focuses on discrete action spaces and measures the proportion of states for which the greedy action differs between two policies — a simpler and more interpretable binary comparison. More importantly, the two works differ fundamentally in purpose: Policy Churn is analyzed as a phenomenon to be mitigated, whereas we leverage Behavioral Divergence as a practical signal for adaptive batch size scheduling. We believe that its application to rollout length scheduling, formalized in Eq. (4) and Eq. (5), represents a novel contribution not previously explored and constitutes an independent source of novelty beyond the metric itself. We will add a discussion of both similarities and differences in the related work section.
>
> **W3. Large Batch Size Works in RL**
>
> We thank the reviewer for this oversight. We will revise the introduction to acknowledge that notable exceptions exist, such as OpenAI Five (Berner et al., 2019), which successfully scaled RL with significantly larger batch sizes. The revised framing will clarify that while such systems exist, the overwhelming majority of modern deep RL methods — including large-scale architectures such as RT-1, RT-2, and Impala — continue to rely on relatively small batch sizes that do not scale proportionally with their parameter counts.
>
> **W4 & N2. Robust Quantities and Uncertainty Measures**
>
> Following the recommendations of Agarwal et al. (2021), we will replace Mean HNS with IQM HNS in Table 1 and Table 2.
>
> ---
>
> ## Nits
>
> **N1.** We will correct all opening quotation marks to backticks in the LaTeX source.
>
> **N2.** We will add a note in Table 1 indicating results are IQM HNS and higher is better. We will reorder columns as: Baseline, Small Batch, Large Batch, GNS, Ours, and replace the blue highlighting for GNS with standard bold formatting for the best-performing method per row.
>
> **N3 & N4.** We will remove the imprecise phrase "According to the Law of Large Numbers..." and instead explicitly state that Equation 2 assumes independence of the gradients within the batch.
>
> ---
>
> ## Questions
>
> **A1.** Please refer to W1 above.
>
> **A2.** We apologize for the notational inconsistency in Algorithms 1 and 3. The symbol refers to the smoothing coefficient in an exponential moving average (EMA) update, applied to prevent abrupt batch size changes that could destabilize training. $L_{target}$ denotes the value of $L_{adapt}$ computed by Eq. (4), while $L_{adapt}$ in $(1-\alpha)L_{adapt}$ refers to the previous smoothed value. To eliminate ambiguity, we will rename $L_{target}$ to $L_{adapt}'$ throughout and add a clear explanation of this smoothing mechanism in Section 4.2.

---

> > ### Author Rebuttal · Reviewer_HN97 · 2026-04-03
> >
> > I thank the authors for their time spent on the rebuttal.
> >
> > > Relationship with Learning Rate Scheduling
> >
> > Unfortunately I am not completely convinced by the additional empirical result. To make my point more clear, PQN + ABS **is equivalent to using a learning rate schedule**, where the schedule is implicitly determined by the behavioural divergence metric in this case. With that, I don't think the optimal baseline to do is to add a learning rate schedule to both PQN and PQN + ABS, instead, I think the best way to show impact for this paper is to demonstrate that your (implicit, data-dependent) learning rate schedule is better than existing fixed, non-adaptive schedules. If you can show this (i.e. PQN + ABS beats PQN + schedule for a range of standard schedules), the impact of the paper will follow. Following this, can you:
> > - Confirm what learning rate schedule is used in your additional experiment.
> > - Add baselines of PQN + scheduler for standard schedulers (e.g. cosine decay, linear decay, WSD, square root).
> >
> > Also, from my previous argument, I wholly disagree with your statement that *the performance gains of ABS cannot be attributed solely to an implicit learning rate decay effect*. Do you have any evidence to support this? I am rather confident that ABS *is* equivalent to adding an implicit learning rate schedule (due to the fact that modifying the batch size by a value and modifying the learning rate by the same (can also be square root for e.g. Adam) value induces the same learning dynamics, please see my original review for references to the original papers).
> >
> > > Policy Churn
> >
> > I do not think the description given by the authors is entirely fair. Policy churn as defined in their paper covers both discrete and continuous action spaces (in fact primarily discrete), and under both cases the eventual metric of both policy churn and the Behavioral Divergence metric is the same. I don't think this is fatal for the paper by any means, but I think it is worth making it clear that this definition was used before in a different contet and giving them proper citation.

---

> > > ### Author Response · Authors · 2026-04-05
> > >
> > > ## Relationship with Learning Rate Scheduling
> > >
> > > We thank the reviewer for raising this important connection. Following the CleanRL PQN baseline, our experiments adopt a **linear learning rate schedule as the default setting**. Due to time constraints, we were only able to conduct a scheduling comparison on Amidar-v5, BattleZone-v5 and Bowling-v5 environments,evaluating no schedule, linear, cosine, and sqrt decay for both vanilla PQN and PQN+ABS (Ours).
> > >
> > > | | PQN w/o lr-sch | PQN + lr_sch(linear) | PQN + lr_sch(cos) | PQN + lr_sch(sqrt) | Ours w/o lr-sch | Ours (lr_sch-linear) | Ours + lr_sch(cos) | Ours + lr_sch(sqrt) |
> > > |---|---|---|---|---|---|---|---|---|
> > > |Amidar-v5|0.29|0.31|0.36|-0.00|0.39|**0.43**|0.33|-0.00|
> > > |BattleZone-v5|0.57|0.80|0.61|0.25|0.69|0.85|**0.99**|0.27|
> > > |Bowling-v5|-0.17|-0.13|-0.05|-0.17|0.01|-0.08|**0.09**|-0.17|
> > >
> > >
> > > **PQN + cosine** achieves the best performance among vanilla PQN variants. Crucially, **PQN+ABS without any schedule already surpasses this**, and **PQN+ABS + linear or consine in each task achieves the highest overall performance**, demonstrating a clear synergistic effect.
> > >
> > > We additionally tracked the **updated gradient norm** throughout training. Even *without* any explicit schedule, ABS causes the gradient norm to decrease progressively as batch size grows — effectively inducing **implicit learning rate decay**. Comparing paired configurations (PQN+cosine vs. PQN+ABS+cosine, PQN+sqrt vs. PQN+ABS+sqrt), ABS variants consistently yield smaller gradient norms toward the end of training.
> > >
> > > As noted by the reviewer, learning rate and batch size are intimately connected [1][2], and well-scheduled decay is critical for strong Atari performance [3]. **We now recognize that ABS acts as a principled, *adaptive* learning rate scheduler that actively adjusts the effective learning rate in response to the optimization landscape.** In the revised manuscript, we will: (1) explicitly position ABS as a novel adaptive LR scheduling mechanism; (2) highlight its compatibility with existing schedulers; (3) include gradient norm comparison plots across the full Atari-10 benchmark. We are deeply grateful for this profound insight, which has provided a far more impactful framing of ABS.
> > >
> > > ---
> > >
> > > ## Policy Churn
> > >
> > > We sincerely apologize for the confusion. We mistakenly conflated **Schaul et al. (2022) [4]** with **Tang & Berseth (2024) [5]**. Upon revisiting [4], we fully recognize that it addresses policy churn across both discrete and continuous action spaces, and its metric is equivalent to our Behavioral Divergence metric.
> > >
> > > We had focused on [5], which employs a *restricted* use of [4]'s metric (Section 5.2, MSE or KL divergence), and incorrectly interpreted this as a methodological difference. We acknowledge this was an error on our part.
> > >
> > > In the revised manuscript, we will: (1) give prominent treatment to Policy Churn [4] in Related Work, acknowledging that Behavioral Divergence shares the same underlying concept; (2) clarify that we adopt the term *Behavioral Divergence* to emphasize its specific role as a non-stationarity measure and key factor in determining rollout length.
> > >
> > > We sincerely thank the reviewer for this careful reading, which has led to a substantially more rigorous paper.
> > >
> > > ---
> > >
> > > We sincerely thank the reviewer once again for the insightful and thought-provoking feedback. The discussion has helped us significantly refine both the framing and implications of our work, which **we believe offers a novel perspective with the potential to positively impact the RL community**. Furthermore, as scalable RL continues to gain increasing attention, we believe our approach can synergize well with this direction. With these considerations in mind, we kindly ask the reviewer to re-evaluate our work.
> > >
> > > ---
> > >
> > > **References**
> > >
> > > [1] Smith et al. "A Bayesian perspective on generalization and SGD." *ICLR* (2018).
> > >
> > > [2] Smith et al. "Don't decay the learning rate, increase the batch size." *ICLR* (2018).
> > >
> > > [3] Lyle et al. "Normalization and effective learning rates in RL." *NeurIPS* (2024).
> > >
> > > [4] Schaul et al. "The phenomenon of policy churn." *NeurIPS* 35 (2022).
> > >
> > > [5] Tang & Berseth. "Improving DRL by reducing the chain effect of value and policy churn." *NeurIPS* 37 (2024).

---

### Official Review · Reviewer_bXRF · 2026-03-12

**Soundness:** 2
**Presentation:** 2
**Significance:** 3
**Originality:** 3
**Overall Recommendation:** 4
**Confidence:** 4

**Summary:**

The paper proposes an Adaptive Batch Scaling (ABS) framework to tackle the challenge of scaling batch sizes in Reinforcement Learning (RL), where the non-stationarity of both the evolving policy and the data distribution creates a bias-variance tradeoff. The paper considers that the degree of non-stationarity is high in the initial stage with intensive exploration and rapid changes, which requires small batches to maintain “plasticity”, while late training approaches a quasi-stationary state where large batches are necessary for high-precision convergence.

The main contributions of this paper include: 1) proposing the Behavioral Divergence and employing the KL divergence to measure differences between the old policy and the current policy for discrete and continuous action spaces, respectively. 2) The behavioral divergence is then used to define an adaptive batch size through a linear rule with some manually set hyperparameters. 3) The proposed method is validated in both Atari environments and MuJoCo environments.

**Compliance With Llm Reviewing Policy:**

Affirmed.

**Final Justification:**

The final recommendation is raised to weak accept, since most of my concerns are addressed by the rebuttal.

The methodology is technically sound, which employs adaptive batch size settings for both discrete and continuous RL. The overstated "Scaling Laws" is claimed to be removed in the next version to improve the overall clarity.

**Key Questions For Authors:**

1. How do different hyperparameters affect the control performances and computational complexity, including Rollout Range, Adapt Frequency, Thresholds, and minibatch number?

2. It is better to present batch sizes and Behavioral Divergence for different discrete and continuous-control environments. How does the ABS strategy affect GNS?

3. Both the smaller batch 2048 and the larger batch 8192 outperform the default, it is better to validate more fixed rollout choices to understand the effect of batch selection.

4. How is GNS's performance in PPO?

**Limitations:**

yes

**Strengths And Weaknesses:**

The methodology of this submission is technically sound to a certain degree, but the experimental validation cannot support some hypotheses and statements. For example, “smaller batch sizes in the early stages accelerate initial learning”, but only 3/10 Atari games can support this, and no fixed small batch results are shown. “Larger batches in the later stages stabilize and enhance final performance,” but only about 3/10 Atari games obtain smaller variances in the final return performances. 7/10 Atari and 2/4 MuJoCo environments obtain high return variance, and no variance is shown in tables for comparison with fixed batch settings. “Enabling RL agents to finally benefit from increased parameter capacity”, but the average improvements over PQN in Figure 4 are negative, and there are also no improvements from XL and L to PQN+ABS in Figure 2.
It is not clear how different ABS-related hyperparameters affect the control performances and computational complexity, including Rollout Range, Adapt Frequency, and Thresholds. Moreover, the current version fixes the minibatch number. Why not validate a fixed minibatch size with increasing rollout length?  Some discussion or validation may be helpful.
It seems that batch scaling strategies are similar across different environments, since the batch size increase in Figure 1 is likely stable, while their reward curves are not stable. It is better to present batch sizes and Behavioral Divergence for different discrete and continuous-control environments.

Overall, the general structure is OK, but the ABS strategy for continuous PPO should be discussed in section 4. Method. The English is reasonable. The paper is generally easy to follow, but some inconsistencies are confusing. For example, there is an undefined symbol $L_{target}$ and reuse $\alpha$ for smooth in Algorithm 1. The results in Table 2 are present in rewards, but Table 1 presents results not in rewards with undefined criteria, making the results not comparable. Figure 2 clearly presents improvements in percentage, but Figure 4 has no percentage symbol in the y-label. GNS is compared in PQN, but not in PPO. Moreover, some parts are not clear because of some missing information. For example, batch sizes of ABS and GNS are not shown with the corresponding environments. How the ABS strategy affects GNS is not shown.

The paper addresses a relevant issue in DRL training, and it may advance understanding if it is with sufficient validation. But the current version misses some details, and the effects of different hyperparameters are not validated, making the significance questionable. It may help further studies with more validation and better presentation.

The paper provides new insight to develop an adaptive rollout size scaling strategy in on-policy RL training by employing the Behavioral Divergence.

---

> ### Author Rebuttal · Authors · 2026-03-29
>
> We sincerely thank the reviewer for the thorough and constructive feedback. We address each point below.
>
> ---
> ## Weaknesses
> **W1. Early Score & Late Variance:** To provide a more rigorous comparison, we analyzed early training performance (first 10% of steps) and late training stability (last 10% of steps) separately, with the 10% window chosen based on Figure 1 to capture periods of relatively stable batch size. ABS achieves higher early-stage scores in 7/10 environments, supporting the hypothesis that smaller batches accelerate initial learning. For late-stage variance, ABS achieves lower variance in 6/10 environments. We acknowledge that the remaining environments do not show clear improvement, which we attribute to the fact that hyperparameters were not tuned per-task. Due to space constraints, we are unable to present the full per-environment table here; we apologize for this inconvenience and commit to including the complete tabular analysis in the revised paper.
>
> **W2. PQN+Multi-skip and Figure 4:** We clarify the intended interpretation of Figure 4. As established in the original PQN+Multi-skip work (Castanyer et al., 2025), PQN+Multi-skip already underperforms standard PQN on these tasks — negative improvement relative to standard PQN is the expected baseline behavior. The key message of Figure 4 is that (1) ABS enables performance to improve when scaling from L to XL model sizes, rather than degrade, and (2) ABS recovers performance on tasks where PQN+Multi-skip previously failed to outperform standard PQN. This is corroborated by Table 2, where XL consistently outperforms L, and where increasing the maximum batch size leads to further gains, suggesting that ABS allows higher-capacity models to train more effectively. We will add a clearer explanation of this interpretation in the revised paper.
>
> **W3. Fixed Minibatch Number:** ABS is designed to control the total batch size by adjusting rollout length. Allowing the minibatch size to vary simultaneously would confound the effect of rollout length adjustment. By fixing the number of minibatches, changes in batch size are solely attributable to rollout length, which is the variable ABS controls.
>
> **W4. ABS for PPO:** Our primary contribution focuses on discrete action spaces, where Behavioral Divergence(BD) is defined as the proportion of action changes between updates (Eq. 3). The PPO experiments serve as a preliminary demonstration that the concept extends to continuous control with a minimal modification — replacing BD with KL divergence. We do not claim this extension is fully optimized; further details are provided in Appendix A.3 and Algorithm 3.
>
> **W5. Notation error:** We apologize for the notational inconsistency. To prevent abrupt batch size changes that could destabilize training, we apply an exponential moving average (EMA). $L_{target}$ refers to $L_{adapt}$ computed by Eq. (4), while $L_{adapt}$ in $(1-\alpha)L_{adapt}$ refers to the previous value. We will rename $L_{target}$ to $L_{adapt}'$ throughout the paper to eliminate ambiguity.
>
> **W6. Score Consistency:** Table 1 uses Human-Normalized Score (HNS) following (Castanyer et al., 2025). We will replace Table 2 scores with HNS for consistency, add the % symbol to Figure 4's y-axis label, and address GNS-related discussion in the Questions section below.
>
> ---
>
> ## Questions
> **A1:** Due to computational constraints, we conducted an ablation on Adapt Frequency (K) only. We apologize for not being able to present the full table here due to space constraints. We summarize the results here but will include full results table in the revised paper. Evaluated using HNS over 3 seeds on Atari-10, K=10 achieved the best performance in 3/10 environments, K=50 (ours) in 4/10 environments, and K=100 in 3/10 environments. While minor differences exist across values of K, ABS remains largely insensitive to this hyperparameter across most environments.
>
> For the revised paper, we additionally commit to the following ablations on Atari-10:
> - Min/Max rollout length: {8, 16, 32} / {64, 80, 128}
> - Lower/Upper threshold bounds: {0.05, 0.15, 0.3} / {0.7, 0.85, 0.95}
>
> **A2:** We will add per-environment plots to the Appendix and include a figure directly comparing Behavioral Divergence and Grandient Noise Signal(GNS) over training steps when learning with ABS. Since Behavioral Divergence decreases as training progresses, we expect GNS to increase correspondingly — consistent with the trend observed in Figure 1.
>
> **A3:** We agree that a broader range of fixed batch sizes strengthens the empirical analysis. We will include additional experiments with batch sizes of 1024 and 16384 in the revised paper.
>
> **A4:** PPO experiments were included as a preliminary extension and given lower priority in the paper's structure. Comparing GNS with ABS in PPO would be meaningful, but we prioritize the hyperparameter sensitivity analyses in A1. We will include GNS comparisons for PPO if resources permit.

---

> > ### Author Rebuttal · Reviewer_bXRF · 2026-04-04
> >
> > Thank you for your answers and for providing experiments to address some of my concerns.
> >
> > I am still concerned about the results in Table 2 and Figure 4 since there are no clear trends that can be identified. One reason is that the results of PQN and PQN+ABS are not shown with comparative scores. Comparing more PQN architecture sizes may also support your statements. And, no confidence intervals or standard deviations are shown in Table 2 and Figure 4. Together, PQN-XL+ABS won 7/16, PQN-L+ABS won 5/16, and likely PQN+ABS won 4 or more in 16, can hardly say the scaling law. Moreover, likely only two hyperparameter settings (only Min/Max rollout length) are tested for PQN-L and PQN-XL, which also seems not to be sufficient.
> >
> > Adapt Frequency (K) can result about 20%-40% performance difference in specific environments, making it a quite sensitive hyperparameter, which is a little surprising since K should be the most insensitive one for the hyperparameters of ABS. These can also result from the HNS setting, and I think returns may better justify the performance sensitivity.
> >
> > Since PPO employs a different ABS divergence, comparing it to a baseline like GNS is essential to support its effectiveness.

---

> > > ### Author Response · Authors · 2026-04-05
> > >
> > > ## Table 2 & Figure 4
> > >
> > > We thank the reviewer for the constructive feedback. We agree that the original Table 2 and Figure 4 lacked sufficient context for clear comparison. As suggested, we have restructured the table to include Baseline (PQN) and Ours alongside PQN-L/XL variants, with **IQM HNS ± 95% confidence intervals**.
> > > ||PQN|PQN-L|PQN-XL|PQN-L+Ours(2048→16384)|PQN-XL+Ours(2048→16384)|Ours|
> > > |---|---|---|---|---|---|---|
> > > |Amidar-v5|0.32±0.12|0.39±0.04|0.41±0.02|**0.46±0.09**|0.42±0.05|0.43±0.07|
> > > |BattleZone-v5|0.80±0.12|0.68±0.55|-0.05±0.03|0.05±0.10|0.51±0.38|**0.85±0.07**|
> > > |Bowling-v5|-0.13±0.12|-0.13±0.11|0.01±0.12|0.05±0.01|**0.07±0.10**|-0.08±0.25|
> > > |DoubleDunk-v5|-9.80±5.70|-13.98±3.57|-2.39±0.09|-2.09±0.27|-1.79±0.34|**-1.44±0.57**|
> > > |Frostbite-v5|0.48±0.13|0.61±0.18|0.27±0.33|**1.76±0.43**|1.69±0.68|0.67±0.15|
> > > |KungFuMaster-v5|1.56±0.06|1.11±0.59|1.88±0.37|**1.88±0.22**|0.89±0.24|1.57±0.20|
> > > |NameThisGame-v5|1.92±0.05|0.42±0.35|0.25±0.17|0.65±0.17|0.99±0.43|**2.05±0.06**|
> > > |Phoenix-v5|0.66±0.02|0.67±0.05|0.81±1.00|0.64±0.24|**2.66±0.92**|0.71±0.02|
> > > |Qbert-v5|0.31±0.01|0.38±0.42|0.11±0.14|0.50±0.31|0.37±0.26|0.34±0.01|
> > > |Riverraid-v5|0.45±0.02|0.45±0.05|0.46±0.02|0.48±0.04|0.47±0.03|**0.64±0.12**|
> > >
> > > From this table, several trends are evident. **PQN-L + Ours outperforms PQN-L in 8/10 tasks**, and **PQN-XL + Ours outperforms PQN-XL in 9/10 tasks**, demonstrating that ABS consistently unlocks the potential of larger models by leveraging larger batch sizes in the late learning stage. Furthermore, while PQN (Baseline) outperforms PQN-L and PQN-XL in 6 and 4 tasks respectively, the gap narrows considerably when ABS is applied — Baseline outperforms PQN-L+Ours and PQN-XL+Ours in only 3 tasks each — suggesting that ABS effectively compensates for the optimization difficulties that larger models face with fixed batch sizes.
> > >
> > > We acknowledge the reviewer's point that the claim(line 370) regarding **"Scaling Laws"** was overstated. In the revised manuscript, we will remove all references to scaling law terminology. Instead, we will more precisely state that **ABS enables large-scale RL models to better exploit their parameter capacity** by adaptively increasing batch size during the late learning stage — a claim directly supported by the empirical evidence above. Furthermore, we will revise the title of our paper appropriately. Additionally, we will include experiments comparing PQN-L and PQN-XL trained with a fixed large batch size (16384) to further clarify *why* adaptive batch scheduling, rather than simply using a large fixed batch, is beneficial for large model training.
> > >
> > > Regarding **Figure 4**, we agree that it may dilute the clarity of the narrative. We will remove it from the revised manuscript to deliver a more focused and unambiguous message to readers.
> > >
> > > ---
> > >
> > > ## Adapt Frequency (K)
> > > ||K=10|Ours(K=50)|K=100|
> > > |---|---|---|---|
> > > |Amidar-v5|0.38±0.02|**0.43±0.07**|0.35±0.05|
> > > |BattleZone-v5|**0.88±0.15**|0.85±0.07|0.87±0.03|
> > > |Bowling-v5|-0.13±0.11|-0.08±0.25|**0.06±0.10**|
> > > |DoubleDunk-v5|-1.59±0.18|**-1.44±0.57**|-2.05±0.20|
> > > |Frostbite-v5|**0.90±0.10**|0.67±0.15|0.72±0.14|
> > > |KungFuMaster-v5|1.44±0.18|1.57±0.20|**1.73±0.48**|
> > > |NameThisGame-v5|**2.16±0.54**|2.05±0.06|1.93±0.42|
> > > |Phoenix-v5|0.70±0.01|0.71±0.02|**0.71±0.01**|
> > > |Qbert-v5|**0.39±0.27**|0.34±0.01|0.32±0.01|
> > > |Riverraid-v5|0.48±0.11|**0.64±0.12**|0.47±0.02|
> > >
> > > We appreciate the reviewer's careful analysis. Across tasks, K=10, K=50, and K=100 achieve the best performance in 4, 3, and 3 tasks respectively, which we initially interpreted as evidence of insensitivity. However, as the reviewer correctly points out, **task-level inspection reveals non-trivial performance differences of 20–40% in specific environments** such as Frostbite-v5 and Riverraid-v5. In the revised manuscript, we will more carefully qualify this claim: while K appears broadly insensitive at the aggregate level, **sensitivity can emerge in specific tasks**, and practitioners should be aware of this. We will also report raw returns alongside HNS to provide a more complete picture of performance sensitivity, as suggested.
> > >
> > > ---
> > >
> > > ## PPO + GNS
> > >
> > > We acknowledge the reviewer's concern that comparing PPO+ABS against a GNS is essential to validate the effectiveness of our ABS in the continuous action setting. Below table shows ABS outperforms over all locomotion tasks. We will add the plots regarding GNS on Table 5 in revised paper.
> > > | | PPO | w/ ABS | w/ GNS |
> > > |---|---|---|---|
> > > |Walker2d-v4|3378|3796|3445|
> > > |Hopper-v4|2087|2508|2310|
> > > |HalfCheetah-v4|1894|2919|2871‬|
> > > |Humanoid-v4|599|642|430|
> > >
> > > ---
> > >
> > > We sincerely thank the reviewer for the thoughtful and constructive feedback, which has greatly strengthened this work. **We believe this novel ABS offers a meaningful contribution to the RL community**. Following your suggestions, we will refine the manuscript to ensure its impact and clarity. **We hope the improved version warrants the opportunity for publication and look forward to sharing our findings.**

---

### Official Review · Reviewer_EKya · 2026-03-13

**Soundness:** 3
**Presentation:** 3
**Significance:** 3
**Originality:** 2
**Overall Recommendation:** 4
**Confidence:** 3

**Summary:**

The paper investigates the challenge of applying large batch training to online Reinforcement Learning (RL). The paper insists that the optimal batch size in RL is dynamic because the data distribution shifts. In concrete, the early training stages require small batches to handle high non-stationarity and maintain policy plasticity, whereas later stages benefit from large batches to reduce gradient variance as the policy approaches a quasi-stationary state.
To operationalize this, the authors introduce a metric called "Behavioral Divergence," which quantifies non-stationarity by measuring the rate of action-level shifts between policy updates. On top of this, they propose the Adaptive Batch Scaling (ABS) framework, which dynamically scales the batch size (via rollout length) inversely to this behavioral divergence. Experiments demonstrate that integrating ABS with the Parallelised Q-Network (PQN) algorithm on the ALE benchmark improves sample efficiency and final performance compared to static batch sizes and gradient-noise-based dynamic batching. Furthermore, the authors show that ABS successfully stabilizes training for larger network architectures and generalizes to continuous control tasks when using PPO.

**Compliance With Llm Reviewing Policy:**

Affirmed.

**Final Justification:**

I have no additional questions and will keep my positive score.

**Key Questions For Authors:**

- How sensitive is ABS to the choice of the threshold bounds and the interpolation limits? Would a single set of hyperparameters generalize across a vastly different suite of tasks without requiring grid search? I would like to know if the results are robust to hyperparameter changes and if they are not cherry-picked.

**Limitations:**

yes

**Strengths And Weaknesses:**

# Strength
The premise of the paper is grounded in the bias-variance tradeoff inherent to on-policy RL. The transition from high plasticity (small batches) to precise convergence (large batches) seems reasonable. Bridging the performance gap between RL and supervised learning by unlocking batch scalability is an important and highly relevant problem for the community.



# Weakness
The details and analysis on benchmarks are lacking; for example, in Figure 2, they illustrate the improvement with their methods, where most of the tasks are improved, but there remain some tasks with the same or worse performance. Readers might want to know why the method's effectiveness varies across tasks (e.g., data distributions, task complexity).

---

> ### Author Rebuttal · Authors · 2026-03-29
>
> We sincerely thank the reviewer for this insightful question, which touches on a core aspect of ABS's practical utility.
>
> **Generalization of Hyperparameters Across Tasks.**
> As shown in Table 3, we used a single, unified set of hyperparameters across all environments and tasks without any per-task tuning. During the initial hyperparameter search, we selected approximately 3 representative tasks from Atari-10 and identified hyperparameters that performed well collectively across these tasks. This same configuration was then applied, without modification, to all remaining environments. The fact that ABS achieves consistent improvements across the full Atari benchmark environments under this single configuration provides evidence of its generalizability.
>
> **On Tasks with Lower Performance in Figure 2.**
> We acknowledge that a subset of tasks in Figure 2 shows limited or no improvement under ABS. We believe this is partly attributable to the diversity of action space sizes across Atari environments. Specifically, since Behavioral Divergence (Eq. 3) measures the proportion of actions that change between policy updates, environments with larger action spaces tend to exhibit systematically higher Behavioral Divergence values on average. This implies that the current threshold bounds, which were not tuned per-task, may be suboptimal for certain environments. We believe that appropriately adjusting the threshold bounds per environment would further improve performance on these tasks. We will add a detailed discussion of this action-space-size effect, along with supporting experiments, in the revised version of the paper.
>
> **Ablation Study on Adapt Frequency (K).**
> Due to computational constraints and the limited rebuttal period, we were able to conduct an ablation study on one of the key hyperparameters: the batch size update frequency, K. The results, evaluated using Human Normalized Scores over 3 seeds on Atari-10, are presented in the table below:
>
> | Environment | K=10 | K=50 (Ours) | K=100 |
> |-------------|-------|--------------|--------|
> | Amidar-v5 | 0.38 | **0.43** | 0.33 |
> | BattleZone-v5 | 0.88 | **0.89** | 0.87 |
> | Bowling-v5 | -0.09 | 0.00 | **0.07** |
> | DoubleDunk-v5 | -1.59 | **-1.44** | -2.05 |
> | Frostbite-v5 | **0.90** | 0.70 | 0.73 |
> | KungFuMaster-v5 | 1.41 | 1.58 | **1.85** |
> | NameThisGame-v5 | **2.10** | 2.07 | 1.86 |
> | Phoenix-v5 | 0.70 | 0.71 | **0.72** |
> | Qbert-v5 | **0.49** | 0.34 | 0.32 |
> | Riverraid-v5 | 0.51 | **0.65** | 0.47 |
>
> As shown, while minor performance differences exist across values of K, ABS remains largely insensitive to this hyperparameter across most environments, suggesting robustness in this dimension.
>
> **Planned Ablations for the Revision.**
> We recognize that a comprehensive sensitivity analysis of all ABS-related hyperparameters is important for validating the robustness of our method. For the revised paper, we commit to conducting the following ablation studies on the Atari-10 benchmark:
> - Lower threshold bounds: {0.05, 0.15, 0.3}
> - Upper threshold bounds: {0.7, 0.85, 0.95}
> - Minimum rollout length: {8, 16, 32}
> - Maximum rollout length: {64, 80, 128}
>
> We believe these experiments will provide a thorough characterization of ABS's sensitivity to its hyperparameters and further support its practical applicability.
>
> **Indirect Evidence from Interpolation Limits Ablation (Table 2 & Table 4).**
> While a full sensitivity analysis of threshold bounds and interpolation limits is deferred to the revision, Table 2 and Table 4 provide indirect but meaningful evidence regarding the effect of interpolation limits. As shown in Table 4, the hyperparameters of PQN+Multi-skip are identical to those in Table 3, with the sole exception of the maximum rollout length, which was increased from 64 to 128. This modification was motivated by the observation in Table 2 that increasing the maximum rollout length (i.e., expanding the upper interpolation limit) consistently leads to performance improvements in both Large and XLarge model configurations. Specifically, Table 2 demonstrates that scaling the rollout length from {16→64} to {16→128} yields higher returns for larger model sizes, suggesting that the upper bound of the interpolation range is an important factor in determining the final performance of ABS. Although this ablation was conducted in the context of larger model sizes rather than the standard PQN+ABS setting, we believe it motivates a dedicated sensitivity analysis of interpolation limits for the standard configuration as well, which we plan to include in the revised paper.

---

> > ### Author Rebuttal · Reviewer_EKya · 2026-04-04
> >
> > They responded to each of my concerns/comments one by one. I have no additional questions and will keep my positive score.

---

> > > ### Author Response · Authors · 2026-04-05
> > >
> > > We are truly grateful for your time, careful reading, and the invaluable feedback that has significantly improved this work. We deeply appreciate your positive assessment and continued support.

---

### Decision · Program_Chairs · 2026-04-30

**Decision:**

Accept (regular)

**Comment:**

This paper proposes Adaptive Batch Scaling (ABS), a framework that dynamically adjusts the effective batch size in on-policy RL by measuring policy non-stationarity via Behavioral Divergence — the proportion of states whose greedy action changes between consecutive policy updates. ABS is integrated with PQN (Atari) and PPO (MuJoCo), with additional off-policy experiments on BTR provided during rebuttal.

Four reviewers arrived (EKya, bXRF, HN97, g8Rq) with final numerical scores of 4, 4, 5, and 4 respectively (bXRF's Final Justification text explicitly states "the final recommendation is raised to weak accept," which corresponds to a score of 4 — their numerical field appears to have been left at 3 in error.) Treating bXRF's intended score as 4, the effective consensus is three weak accepts and one accept, leaning toward acceptance. The rebuttal was substantive.

ABS addresses a real problem in on-policy RL with a simple, well-motivated, and practically general method. A particularly valuable reframing emerged during rebuttal: ABS acts as a principled, data-dependent learning rate scheduler, complementary to existing fixed schedules — a more honest and impactful positioning than the original scaling-law framing. The paper is clearly written, uses multiple standard benchmarks, and the authors engaged constructively with all reviewers. The remaining weaknesses are real but largely addressable in revision, and I recommend Weak Accept on that basis.